# Tissue specificity-aware TWAS (TSA-TWAS) framework identifies novel associations with metabolic, immunologic, and virologic traits in HIV-positive adults

**Binglan Li**[1], **Yogasudha Veturi**[2], **Anurag Verma**[2], **Yuki Bradford**[2], **Eric S. Daar**[3], **Roy M. Gulick**[4], **Sharon A. Riddler**[5], **Gregory K. Robbins**[6], **Jeffrey L. Lennox**[7], **David W. Haas**[8,9], **Marylyn D. Ritchie**[2,10]*

1 Department of Biomedical Data Science, Stanford University, Stanford, California, United States of America, 2 Department of Genetics, University of Pennsylvania, Philadelphia, Pennsylvania, United States of America, 3 Lundquist Institute at Harbor-UCLA Medical Center, Torrance, California, United States of America, 4 Weill Cornell Medicine, New York City, New York, United States of America, 5 Department of Medicine, University of Pittsburgh, Pittsburgh, Pennsylvania, United States of America, 6 Division of Infectious Diseases, Massachusetts General Hospital, Boston, Massachusetts, United States of America, 7 Emory University School of Medicine, Atlanta, Georgia, United States of America, 8 Departments of Medicine, Pharmacology, Pathology, Microbiology & Immunology, Vanderbilt University School of Medicine, Nashville, Tennessee, United States of America, 9 Department of Internal Medicine, Meharry Medical College, Nashville, Tennessee, United States of America, 10 Institute for Biomedical Informatics, University of Pennsylvania, Philadelphia, Pennsylvania, United States of America

* marylyn@pennmedicine.upenn.edu

**Data Availability Statement:** All summary data and results are within the manuscript and its Supporting Information files. The individual level genotype and phenotype data is not publicly

## Abstract

As a type of relatively new methodology, the transcriptome-wide association study (TWAS) has gained interest due to capacity for gene-level association testing. However, the development of TWAS has outpaced statistical evaluation of TWAS gene prioritization performance. Current TWAS methods vary in underlying biological assumptions about tissue specificity of transcriptional regulatory mechanisms. In a previous study from our group, this may have affected whether TWAS methods better identified associations in single tissues versus multiple tissues. We therefore designed simulation analyses to examine how the interplay between particular TWAS methods and tissue specificity of gene expression affects power and type I error rates for gene prioritization. We found that cross-tissue identification of expression quantitative trait loci (eQTLs) improved TWAS power. Single-tissue TWAS (i.e., PrediXcan) had robust power to identify genes expressed in single tissues, but, often found significant associations in the wrong tissues as well (therefore had high false positive rates). Cross-tissue TWAS (i.e., UTMOST) had overall equal or greater power and controlled type I error rates for genes expressed in multiple tissues. Based on these simulation results, we applied a tissue specificity-aware TWAS (TSA-TWAS) analytic framework to look for gene-based associations with pre-treatment laboratory values from AIDS Clinical Trial Group (ACTG) studies. We replicated several proof-of-concept transcriptionally regulated gene-trait associations, including *UGT1A1* (encoding bilirubin uridine diphosphate glucuronosyltransferase enzyme) and total bilirubin levels (p = $3.59 \times 10^{-12}$), and *CETP* (cholesteryl ester transfer protein) with high-density lipoprotein cholesterol (p = $4.49 \times 10^{-12}$).

available, but is available through collaboration with the ACTG (https://actgnetwork.org/).

**Funding:** This project was supported by the National Institute of Allergy and Infectious Diseases (NIAID award number U01AI068636), the National Institute of Mental Health, and the National Institute of Dental and Craniofacial Research. Grant support included TR000124 (to E.S.D.); AI077505, TR000445, AI069439 (to D.W.H.); and the National Institute of Allergy and Infectious Disease (NIAID award AI077505 and AI116794 (to M.D.R.). Clinical research sites that participated in ACTG protocols ACTG 384, A5095, A5142, A5202 or A5257, and collected DNA under protocol A5128 were supported by the following grants from the National Institutes of Health (NIH): A1069412, A1069423, A1069424, A1069503, AI025859, AI025868, AI027658, AI027661, AI027666, AI027675, AI032782, AI034853, AI038858, AI045008, AI046370, AI046376, AI050409, AI050410, AI050410, AI058740, AI060354, AI068636, AI069412, AI069415, AI069418, AI069419, AI069423, AI069424, AI069428, AI069432, AI069432, AI069434, AI069439, AI069447, AI069450, AI069452, AI069465, AI069467, AI069470, AI069471, AI069472, AI069474, AI069477, AI069481, AI069484, AI069494, AI069495, AI069496, AI069501, AI069501, AI069502, AI069503, AI069511, AI069513, AI069532, AI069534, AI069556, AI072626, AI073961, RR000046, RR000425, RR023561, RR024156, RR024160, RR024996, RR025008, RR025747, RR025777, RR025780, TR000004, TR000058, TR000124, TR000170, TR000439, TR000445, TR000457, TR001079, TR001082, TR001111, and TR024160. NIH: https://www.nih.gov/ NIAID: https://www.niaid.nih.gov/ The funders had no role in study design, data collection and analysis, decision to publish, or preparation of the manuscript.

**Competing interests:** I have read the journal's policy and the authors of this manuscript have the following competing interests: G.K.R. has been on principal investigator on research grants to Massachusetts General Hospital from Gilead Sciences, Citius Pharmaceuticals, Emergent Biosolutions, Pfizer and Leonard Meron Bioscience -all research support paid to the institution. E.S.D. has received research grant support from Gilead, Merck and ViiV and has been consultant for Abbvie, Gilead, Merck and Genentech. S.A.R. has been an investigator on research grants to the University of Pittsburgh from Gilead, Merck, and Bristol Myers Squibb and she has been a consultant to Novimab. M.D.R. is on the scientific

We also identified several novel genes associated with metabolic and virologic traits, as well as pleiotropic genes that linked plasma viral load, absolute basophil count, and/or triglyceride levels. By highlighting the advantages of different TWAS methods, our simulation study promotes a tissue specificity-aware TWAS analytic framework that revealed novel aspects of HIV-related traits.

## Author summary

Transcriptome-wide association studies (TWAS) are a type of bioinformatics methodology for identifying complex trait-associated genes. There have been various TWAS methods, each developed under distinct biological assumptions of how genes contribute to complex traits. It is unclear, however, how powerful different TWAS methods are under a variety of biological scenarios. Here, we design an unbiased simulation strategy to evaluate the performance of multiple representative TWAS methods. We find that no one method fits all. Different TWAS methods are advantageous at dealing with different biological scenarios and answering different research questions. Thus, we propose a novel TWAS analytic framework that integrates and maximizes the performance of multiple TWAS methods, and validate its capability using a well-studied real-world dataset. In summary, our study provides quantitative evaluation of method performance to aid future TWAS experimental design and understanding of genes underlying complex human traits. The TWAS evaluation tool is made publicly available.

## Introduction

Translating fundamental genetics research discoveries into clinical research and clinical practice is a challenge for biomedical studies of complex human traits [1,2]. Greater than 90% of complex trait-associated single-nucleotide polymorphisms (SNPs) identified via genome-wide association studies (GWAS) are located in noncoding regions of the human genome [3,4]. The difficulty in making connections between noncoding variants and downstream affected genes can hinder the translatability of GWAS discoveries to clinical research. The emerging transcriptome-wide association studies (TWAS) are a type of recently developed bioinformatics methodology that provide a means to address the challenge of GWAS translatability. TWAS mitigates the translational issue by integrating GWAS data with expression quantitative trait loci (eQTLs) information to perform gene-level association analyses. TWAS hypothesizes that SNPs act as eQTLs to collectively moderate the transcriptional activities of genes and thus influence complex traits of interest [5,6]. Accordingly, TWAS methods in general comprise two steps. The first step in TWAS is to impute the genetically regulated gene expression (GReX) for research samples in a tissue-specific manner. The second step is to conduct association analyses between GReX and the trait of interest to evaluate the gene-trait relationship for statistical significance [7–9]. Genome-wide eQTLs data are now available for various primary human tissues (e.g., liver, brain and heart) thanks to large-scale eQTL consortia including the Genotype-Tissue Expression (GTEx) project [10] and the eQTLGEN consortium [11]. The considerable centralized eQTL data have been fostering the development and application of TWAS.

While TWAS is an innovative and potentially powerful computational approach, several factors can influence TWAS. The choice of eQTL datasets matters for the performance of

advisory board for Cipherome and Goldfinch Bio.
All other authors have no conflicts to disclose.

TWAS [12]. Most available eQTLs to date are identified in a tissue-by-tissue manner [5,10]. This approach, however, does not leverage the potential for shared transcriptional regulatory mechanisms across tissues, and can be limited by sample sizes of single tissues. One way to overcome this limitation is to take into consideration all available tissues, so as to increase sample sizes and improve the quality of eQTL datasets. We referred this type of eQTL detection method as the integrative tissue-based eQTL detection method [13–15]. Without a simulation study, however, it was unclear *how the choice of eQTL detection methods will impact TWAS*.

Another prominent question in TWAS studies is the choice of the association approaches. TWAS started with single-tissue association approaches, such as PrediXcan [5] and FUSION [6]. The most recent TWAS methods, such as UTMOST [15] and MulTiXcan [16], perform cross-tissue association analyses. Such TWAS methods evaluate whether a gene is significantly associated with a trait by integrating association data across tissues and adjusting for the statistical correlation structure among tissues. However, genes may vary substantially with regard to how they are regulated across tissues. When a gene is specifically expressed in a single or few tissues versus expressed in multiple tissues, *how will tissue specificity of gene expression affect TWAS power and type I error rates*?

Another appealing feature of TWAS is its capacity for tissue-specific association analyses thanks to the availability of tissue-specific eQTLs in a variety of primary human tissues. However, several recent studies revealed shared regulatory mechanisms across multiple human tissues [17] and showed that *cis*-eQTLs are less tissue-specific than other regulatory elements [10,11]. This suggests that TWAS can possibly identify genes in tissues that share biology with the causal tissue(s), but in fact are not the causal tissues for the trait of interest [18]. While TWAS is likely to identify false positive tissues, to date, the false positive rates of tissues are TWAS is unknown.

The above TWAS challenges can be summarized in two questions—*How does tissue specificity affect TWAS performance*? *How would this impact the choices of TWAS methods*? Available simulation strategies can be limited in answering these questions. Some have not taken into consideration the gene expression correlation structure across tissues [19,20]. Some assume a monogenic structure of transcriptional regulation [13–15,21], rather than the polygenic structure suggested by recent studies [10,22,23]. To address these issues, we applied a novel strategy to simulate eQTLs and gene expression of a wide range of tissue specificity (see **Methods**). We then applied different TWAS methods on the simulated datasets to assess power, type I error rates, and false positive rates of tissues. We found that the tissue specificity affected TWAS performance, with no single type of TWAS method being best for every type of genetic background of transcriptional regulation.

The simulation results motivated the development and implementation of an enhanced, tissue specificity-aware TWAS (TSA-TWAS) analytic framework. We tested the performance of TSA-TWAS analytic framework using AIDS Clinical Trials Group (ACTG) data (described in **Methods**). We showed that the TSA-TWAS was able to both replicate proof-of-concept gene-trait associations and identify novel trait-related genes. The simulation scheme highlighted the effects of tissue specificity on TWAS performance, and that TSA-TWAS could help better understand regulatory mechanisms that underlie complex human traits.

## Results

### Simulation design

We designed a novel simulation framework to investigate how the tissue specificities of eQTLs and gene expression affected TWAS power and type I error rates, and the choices of TWAS methods (Fig 1). We tested two representative eQTL detection methods, elastic net

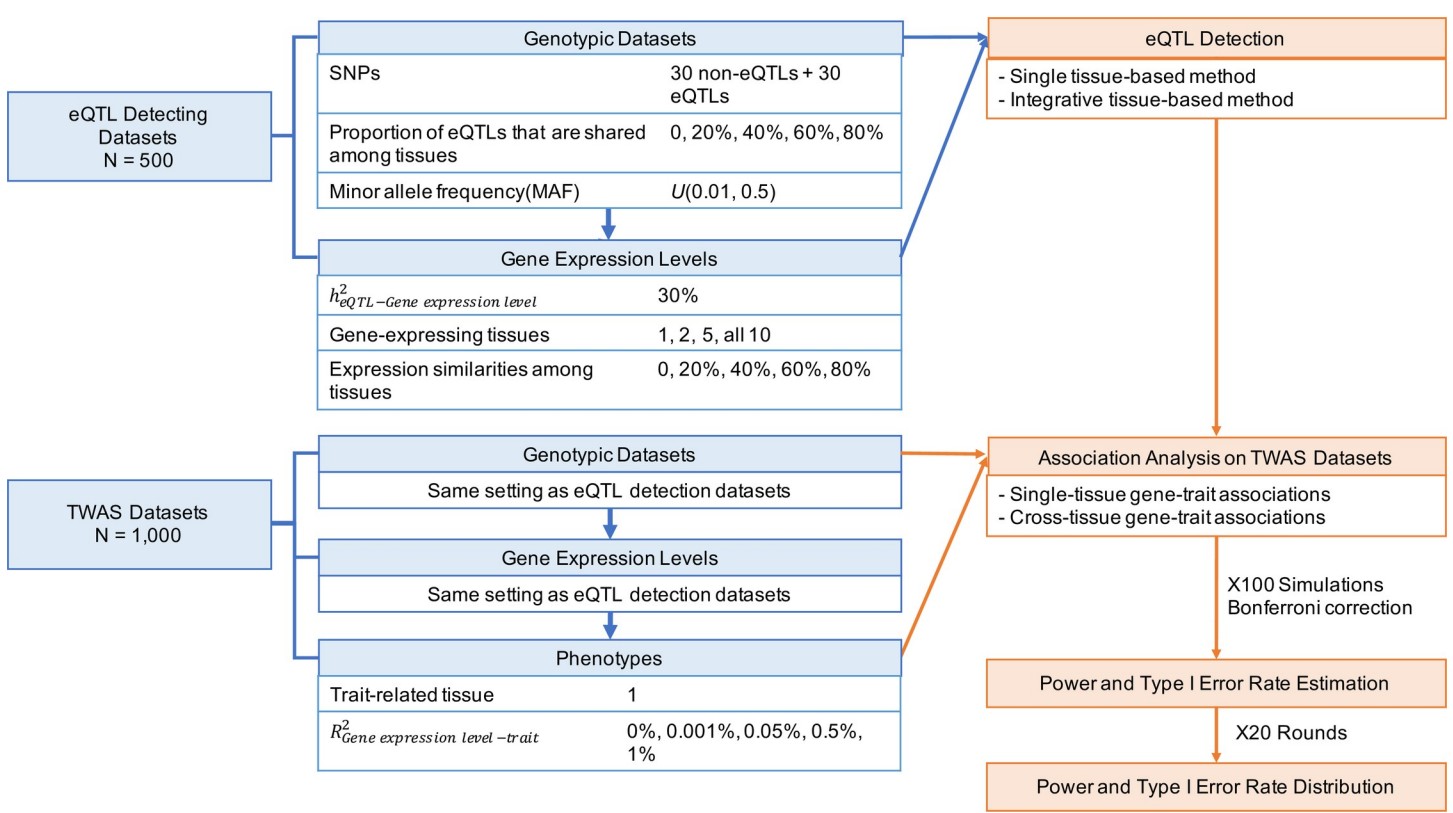

**Fig 1. Cross-tissue TWAS simulation scheme.** With the simulation parameters, we were able to generate SNP-gene-trait relations of varied tissue specificity backgrounds. In each replication, simulated datasets were divided into an eQTL detection dataset and a TWAS dataset. The former was used to identify eQTLs using different eQTL detection methods and the sample size was equivalent to that of GTEx. The detected eQTLs were then passed, separately, to the TWAS dataset to assist gene-level association tests. The TWAS dataset sample size was equivalent of that of the ACTG clinical trial dataset. Two types of gene-level association approaches estimated and ascribed p-values to the simulated gene-trait relations. In each replication, we simulated 100 different SNP-gene-trait pairs for one single point estimation of TWAS gene prioritization performance. All association p-values had been adjusted for the number of genes and tissues in each replication. 20 independent replications were conducted to obtain the distribution of TWAS performance statistics.

(implemented in PrediXcan [5]) and group LASSO (implemented in UTMOST [15]); and two gene-trait association approaches, Principal Component Regression (PC Regression; implemented in MulTiXcan [16]) and Generalized Berk-Jones test (GBJ test; implemented in UTMOST [15]) (Table 1).

Tissue-specific eQTLs were defined as those that were only functioning in one single tissue. Multi-tissue eQTLs were defined as those that had regulatory effect across all gene-expressing tissues (see **Methods**). We generated genes that had different genetic makeup of tissue-specific and multi-tissue eQTLs in a gene to evaluate the influence of tissue specificity of eQTLs on TWAS performance.

**Table 1. TWAS methods tested in this simulation study.**

| eQTL detection methods | | Gene-trait association approaches | | Equivalent developed TWAS methods | PMID |
|---|---|---|---|---|---|
| Type | Name | Type | Name | | |
| Single tissue-based | Elastic net | Single-tissue association | Linear or logistic regression | PrediXcan | 26258848 |
| Integrative tissue-based | Group LASSO | Single-tissue association | Linear or logistic regression | Single-tissue UTMOST | 30804563 |
| Single tissue-based | Elastic net | Cross-tissue association | Principal component regression | MulTiXcan | 30668570 |
| Integrative tissue-based | Group LASSO | Cross-tissue association | Generalized Berk-Jones test | Cross-tissue UTMOST | 30804563 |

Tissue specificity of gene expression was determined by the number of gene-expressing tissues and the similarity of gene expression levels across tissues. (see **Methods**). Tissue-specific genes were those specifically expressed in only one or two tissues. Ubiquitously expressed genes were those expressed in all ten simulated tissues with high gene expression similarity (expression similarity = 60%, 80%). Differentially expressed and similarly expressed genes were those having distinctive gene expression levels (gene expression similarity = 0, 20% 40%) or highly correlated gene expression levels across tissues (gene expression similarity = 60%, 80%), respectively, regardless of the number of gene-expressing tissues. To evaluate the impact of tissue specificity of gene expression on TWAS performance, we generated genes that were expressed in varied numbers of tissues and had diverse gene expression similarities across tissues.

In addition, we designed different strength of gene-trait associations defined by $R^2_{expression-trait}$ (the proportion of phenotypic variation explained by gene expression levels), but the reported results by default were the cases under $R^2_{expression-trait}$ = 1%. Only continuous traits were evaluated in this simulation study, in accordance with ACTG baseline laboratory values in the real-world application dataset.

## Power of different TWAS methods

We did not observe any obvious effect of tissue-specificity of eQTLs on TWAS power with the exception of one condition (Fig 2, bottom row). Specifically, the group LASSO-GBJ test (implemented in UTMOST [15]) had greater power to prioritize genes whose similar gene expression levels were driven by multi-tissue eQTLs (the group LASSO-GBJ test in Fig 2, bottom right) than those whose similar gene expression levels were not driven by multi-tissue eQTLs (the group LASSO-GBJ test in Fig 2, bottom left).

We then asked how eQTL detection methods affected TWAS gene-prioritization power, and whether one eQTL detection method was preferred over another. We found that the integrative tissue-based eQTL detection method had, on average, approximately 2% greater power than the single-tissue method. Take differentially expressed genes for instance, eQTLs identified via the Group LASSO led to 53.8% gene prioritization power of TWAS and eQTLs identified via the Elastic Net led to 50.7% power (Wilcoxon Signed-rank Test p = 5.85×10$^{-4}$; S4 Fig, top right corner). More pairwise comparison results among all TWAS methods can be found in S1 Table. Overall, TWAS gained slightly more power when using eQTLs identified in an integrative tissue context.

Gene-trait association approaches affected TWAS power more so than did choice of eQTL detection method. For tissue-specific genes, SLR consistently had equal or greater power (average 70%) than the cross-tissue association approaches (PC regression and GBJ test; Fig 2, top left triangle). For genes that were expressed in multiple tissues, GBJ test had equal or greater power than SLR (Fig 2, bottom right triangle). Especially for ubiquitously expressed genes, GBJ test had statistically significant greater power (62%) compared to SLR (51%) (Fig 2, bottom right corner, Wilcoxon Signed-rank Test p = 9.4×10$^{-5}$).

The group LASSO-GBJ test (implemented in UTMOST) had a greater power to prioritize genes that had similar gene expression levels across tissues. For genes that were expressed in five tissues, power of the group LASSO-GBJ test increased from 62.2% for differentially expressed genes (Fig 2, top left corner) to 66.6% for similarly expressed gene (Fig 2, bottom right corner). For genes that were expressed in all ten tissues, power of the group LASSO-GBJ test increased from 51.2% for differentially expressed genes (Fig 2, top left corner) to 61.9% for similarly expressed gene (Fig 2, bottom right corner). Moreover, the group LASSO-GBJ test showed equal or statistically significant greater power than other TWAS methods in 65 of the

Proportion of eQTLs that are shared among tissues

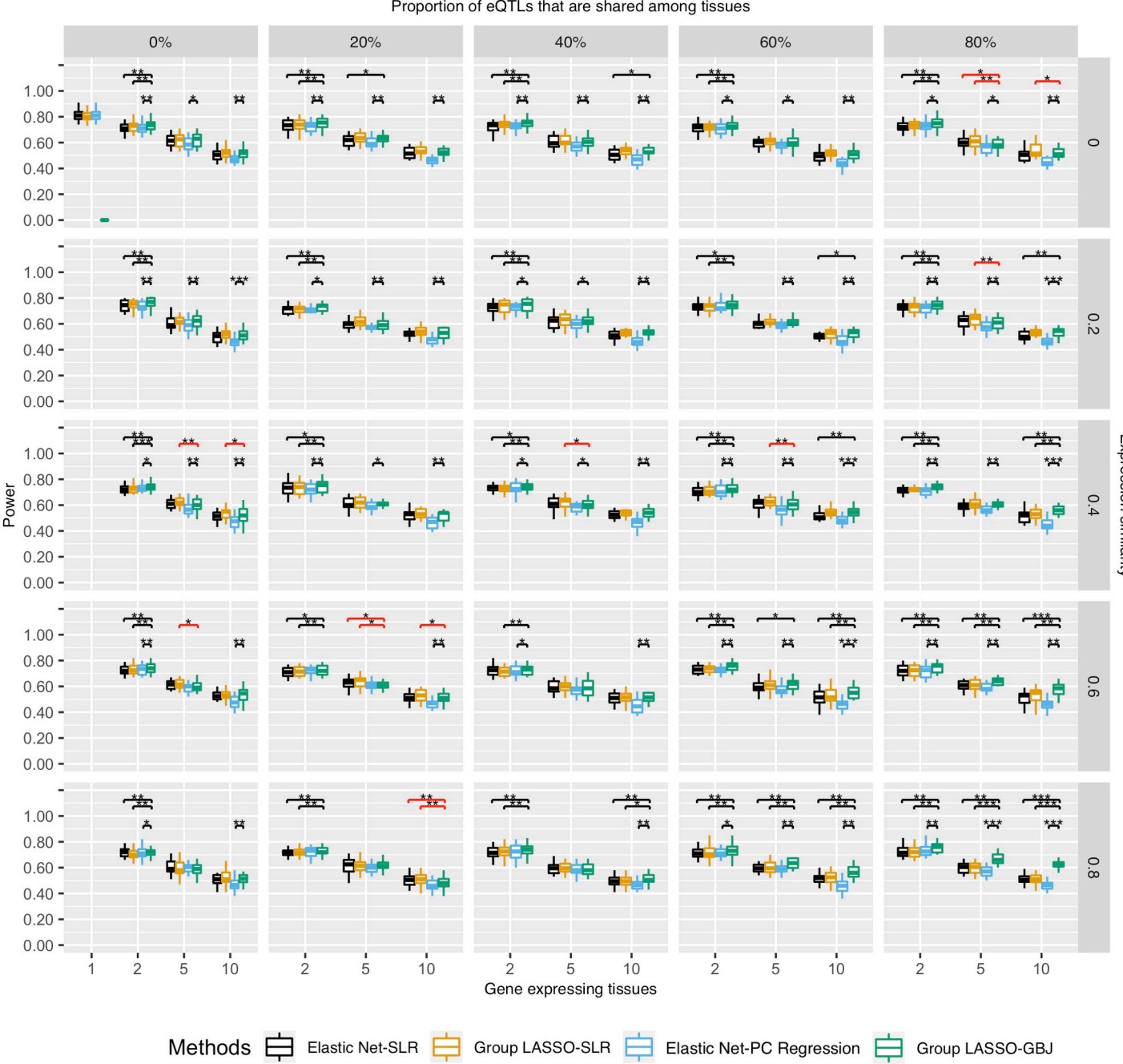

**Fig 2. Power of different TWAS methods in prioritizing genes of varied tissue specificity properties.** Power was the proportion of successfully identified gene-trait associations in the causal tissue out of all simulations. X-axis is the number of gene-expressing tissues. Each column stands for the proportion of eQTLs that are shared among tissues for a gene. Each row is the similarity of gene expression profiles across tissues which is estimated by correlation. Moving from the top left to the bottom right is a gradient spectrum from tissue-specific genes to broadly expressed genes. The colors represent different TWAS methods and y-axis is the power. For tissue-specific genes at the top left, single-tissue TWAS (Elastic Net-SLR) and cross-tissue TWAS (Group LASSO-GBJ) had similar power. For broadly expressed genes at the bottom right, cross-tissue TWAS (Group LASSO-GBJ) had greater power. Brackets showed pairwise comparison of power between the Group LASSO-GBJ and other TWAS methods using Wilcoxon Signed-rank Test. Black brackets were cases where Group LASSO-GBJ had higher power than other three methods; red brackets were cases where Group LASSO-GBJ had lower power than other three methods. *p-value < 0.05, **p-value < 0.01, ***p-value < 0.0001.

76 simulated scenarios (~84%). Black brackets in Fig 2 showed cases where Group LASSO-GBJ had higher power than other three methods; red brackets showed cases where Group LAS-SO-GBJ had lower power than other three methods. Comprehensive statistical test results of power differences are available in S4 Fig and S1 Table. However, GBJ test does not handle the case where the gene was only expressed in one single tissue. This would inevitably lead to greater loss of power when the proportion of tissue-specific genes are higher in a test dataset.

Overall, the group LASSO-GBJ test had equal or greater power in prioritizing genes that were expressed in multiple tissues. Single-tissue association approaches (e.g. SLR) had greater power and robust performance in prioritizing tissue-specific genes.

The strength of gene-trait associations affected TWAS gene prioritization power. The stronger the gene-trait associations, the greater the power for TWAS gene prioritization (Figs 2 and S5–S7).

## Type I error rates of various TWAS methods

All TWAS methods had well-controlled type I error rates ($\leq$ 5%; Fig 3 and S2 Table). Significance thresholds in this simulation were corrected using the Bonferroni approach to control for family-wise error rate. All single-tissue association approaches (Elastic Net-SLR and Group LASSO-SLR) had less type I error rates than the cross-tissue associations approaches (Wilcoxon Signed-rank Test p < 0.01, S8 Fig). Both GBJ test and PC regression had average type I error rates of approximately 5%. The GBJ test showed statistically significant lower type I error rates than PC regression for ubiquitously expressed genes (Wilcoxon Signed-rank p < 0.05, S8 Fig and S2 Table).

## False positives of statistically significant tissues

If not corrected for the number of tested tissues, single-tissue TWAS would have greater power (S9 Fig), but also a higher false positive rate for tissues (S10 Fig). False positive rates of tissues were at least 10% for genes that were expressed in more than one tissue. In effect, while the genes might be related to a trait of interest, 10% of statistically significant results pointed to wrong tissues. The false positive rate of tissues proportionally increased with the number of gene-expressing tissues. The highest false positive rates were seen in the case of ubiquitously expressed genes (S10 Fig, bottom right corner), which on average, had an 84% false positive rate based on 20 random replications. This suggested that any single-tissue TWAS may have 10–84% false positive rate tissues associations if not adjusted for the number of tested tissues.

Adjusting for the number of tested tissues reduced the false positive rates somewhat, but number-wise, the false positive rate may remain quite high. False positive rates of tissues were relatively controlled at approximately 5% for tissue-specific genes (Fig 4, top left corner). False positive rates still increased with the number of tissues in which a gene was expressed (Fig 4). Genes expressed in ten tissues had at least on average a 24% false positive rate. False positive rates were as high as 77% for ubiquitously expressed genes (Fig 4, bottom right corner).

## Validation and support of simulation design

To evaluate whether our simulation findings would translate from in silico parameter designs to real world scenarios, we designed a Monte Carlo simulation process to estimate the trait heritability behind various genetic scenarios (S11 Fig). The results suggested that $R^2_{expression-trait}$ increased with trait heritability (S12 Fig). Heritability of traits with $R^2_{expression-trait}$ = 1% were estimated to be on average 1% (standard error (s.e.) = 0.059%) which were derived from multiple, repeated random sampling. In contrast, the minor allele frequencies (MAF) of eQTLs had

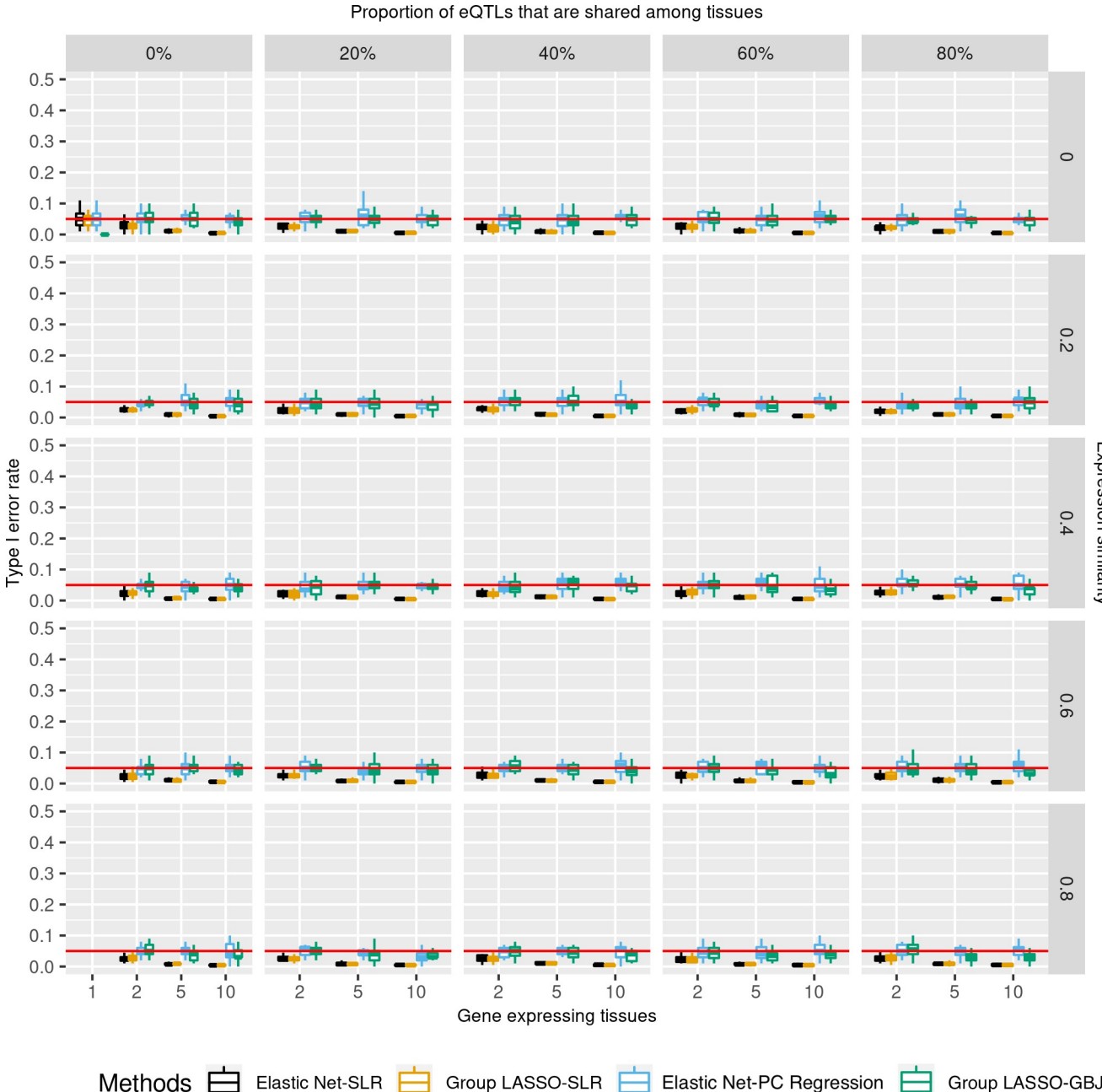

**Fig 3. Type I error rates of different TWAS methods in prioritizing genes of diverse tissue specificity properties.** Type I error rate was the probability that TWAS wrongly identified a gene-trait association as significant while there was not any signal simulated in the dataset. Association p-values were controlled for the number of genes and tested tissues. X-axis is the number of gene-expressing tissues. Each column stands for the proportion of eQTLs that are shared among tissues for a gene. Each row is the similarity of gene expression profiles across tissues which is estimated by correlation. Moving from the top left to the bottom right is a gradient spectrum from tissue-specific genes to broadly expressed genes. The colors represent different TWAS methods and y-axis is the type I error rate. All TWAS methods had controlled type I error rates ($\leq$ 5%).

almost no effect on trait heritability. This suggested that trait heritability positively influenced the strength of gene-trait associations in TWAS. In other words, if a trait was moderated by genetic factors through differential gene expression, the greater a trait's heritability is, the stronger the associations were in TWAS.

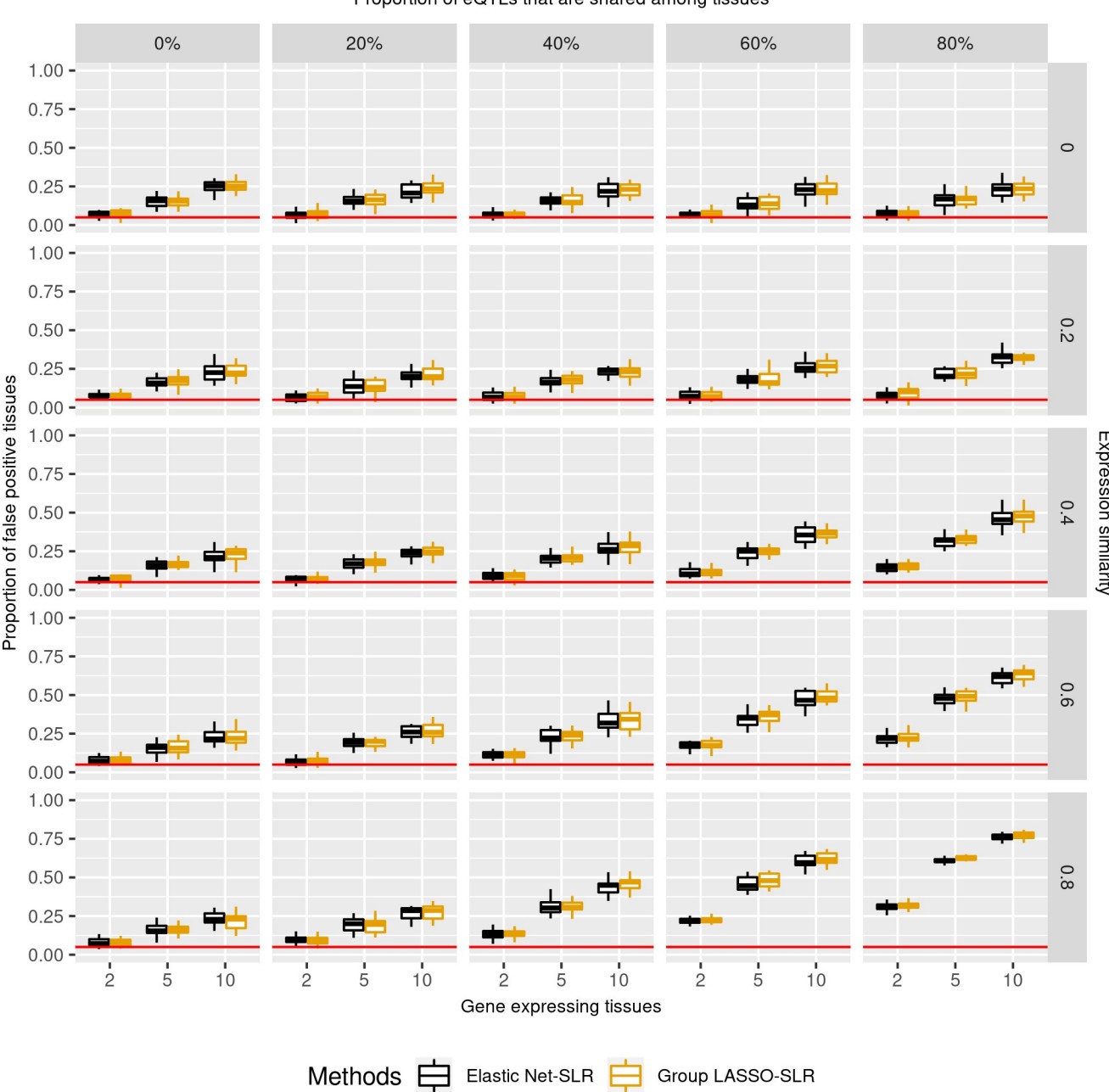

**Fig 4. False positive rates of tissues among statistically significant results.** False positive rates were the proportion of significant associations found in trait-irrelevant tissues amongst all significant results. Association p-values were controlled for the number of genes and tested tissues. X-axis is the number of gene-expressing tissues. Each column stands for the proportion of eQTLs that are shared among tissues for a gene. Each row is the similarity of gene expression profiles across tissues which is estimated by correlation. Moving from the top left to the bottom right is a gradient spectrum from tissue-specific genes to broadly expressed genes. Colors represent different TWAS methods and y-axis is the false positive rate of tissues among statistically significant results. Single-tissue TWAS wrongly identified 5% and 77% trait-irrelevant tissues for tissue-specific genes and broadly expressed genes, respectively.

## Designing the TSA-TWAS analytic framework

Our simulation suggested an influence of tissue specificity on TWAS performance. Thus, we designed a TSA-TWAS analytic framework to balance trade-offs among power and type I error rates (S13 Fig). The idea was illustrated in Fig 5. When trait-related tissue(s) are known,

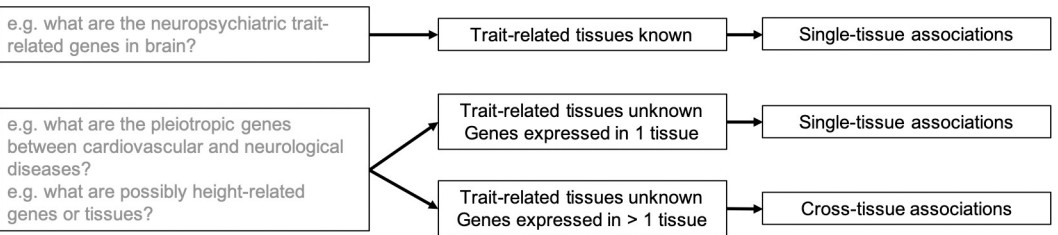

**Fig 5. A proposed TSA-TWAS analytic framework that leverages TWAS performance on genes of different tissue specificity properties.** The framework proposed based on our simulations is as follows: If trait-related tissue(s) are known for a trait or disease of interest, run single-tissue TWAS, for example, PrediXcan. If trait-related tissue(s) are unknown, run cross-tissue TWAS (UTMOST) on the genes that are expressed in more than one tissue and run single-tissue TWAS (PrediXcan) on the genes that are expressed in one single tissue.

we recommend single-tissue TWAS in the known related tissues only. Additionally, we recommend using eQTLs identified by integrative tissue-based eQTL detection methods (for example, group LASSO or MASHR-based eQTL databases), which showed slightly improved power. In contrast, if trait-related tissue(s) are uncertain, it may be better to perform different TWAS analyses for tissue-specific genes and for genes that are expressed in multiple tissues. Single-tissue TWAS will have greater power to identify genes that are expressed in a single tissue. Cross-tissue TWAS will provide overall equal or greater power, as well as controlled type I error rates, for genes that are expressed in multiple tissues.

In real-world, natural data, we can expect a collection of genes that are tissue-specific and another set of genes that are expressed in multiple tissues. We showed that our TSA-TWAS approach had a consistent power of identifying complex-trait related genes in comparison to single-tissue TWAS (SLR) or cross-tissue TWAS (GBJ tests), regardless of the proportion of tissue-specific genes in an analysis (Fig 6 –blue bars). By using the TSA-TWAS framework, the optimal method (Elastic Net-SLR) is used on the genes expressed in a single tissue while simultaneously, the optimal method (Group Lasso-GBJ) is used on the genes expressed in multiple tissues.

## TSA-TWAS replicated known associations

We applied TSA-TWAS to 37 baseline laboratory values from a combined dataset of five clinical trials from AIDS Clinical Trials Group (ACTG) with available genotype data (N = 4,360; Fig 7 and Table 2). We first imputed the GReX to distinguish genes whose GReX were only expressed in one tissue versus multiple tissues. Genes expressed in just one tissue comprised 2,812 (23%) of 12,038 genes on which data were available. The remaining 9,226 (77%) genes had GReX in multiple tissues. Genes expressed in one, and in more than one tissue were tested for associations with baseline laboratory values using single-tissue, and by cross-tissue gene-trait association approaches, respectively (see **Methods**). TSA-TWAS found in total 83 statistically significant gene-trait associations, comprising 45 distinct genes and 10 traits (Fig 8).

We also fine-mapped a credible set of potential trait-related genes (S3 Table). The credible sets added twenty genes that were correlated with the statistically significant signals as a function of LD among SNPs and eQTLs. We further performed colocalization analysis to see if there was supportive evidence to prioritize any of the statistically significant genes. Some of the TSA-TWAS statistically significant genes were supported with a locus regional colocalization probability (locus RCP) > 0.025 (S15 Fig). None of the additional genes that were identified by FOCUS [24] were supported by colocalization analyses.

TSA-TWAS replicated several previously reported risk genes for certain baseline lab values (Table 3). The lowest p-values for association were observed between total plasma

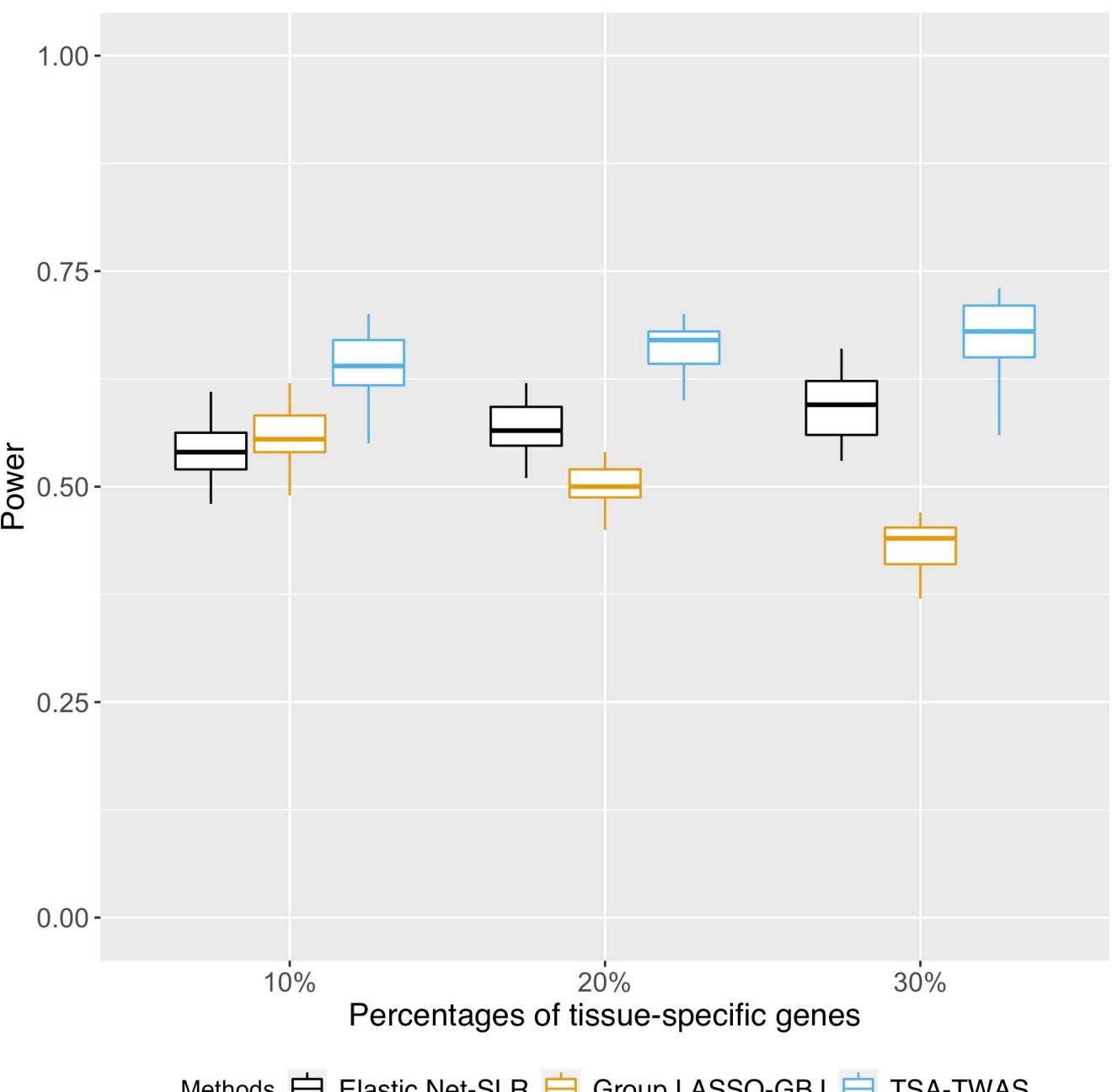

**Fig 6. Power of the TSA-TWAS framework when there were different proportions of tissue-specific genes in the data.** The power of TSA-TWAS was compared to only running single-tissue TWAS (elastic net-SLR) and cross-tissue TWAS (Group LASSO-GBJ test). TSA-TWAS had consistent power of identifying complex trait-related genes and was robust to makeups of tissue-specific and multi-tissue genes in a dataset.

bilirubin levels and several genes on chromosome 2, nearby or overlapping *UGT1A1*. These included *MROH2A* (p = $1.39 \times 10^{-12}$), which has been previously reported by GWAS of various populations [25–28], *UGT1A6* (p = $2.78 \times 10^{-15}$), *UGT1A7* (p = $4.51 \times 10^{-12}$) and *UGT1A1* (p = $3.59 \times 10^{-12}$) [25,26,28,29]. We replicated the well-known association between *CETP* and high-density lipid-cholesterol levels (HDL-c; p = $4.49 \times 10^{-12}$) [30]. Association was also found between *GPLD1* and plasma alkaline phosphatase levels (p = $1.08 \times 10^{-11}$) [31]. *GPLD1* encodes

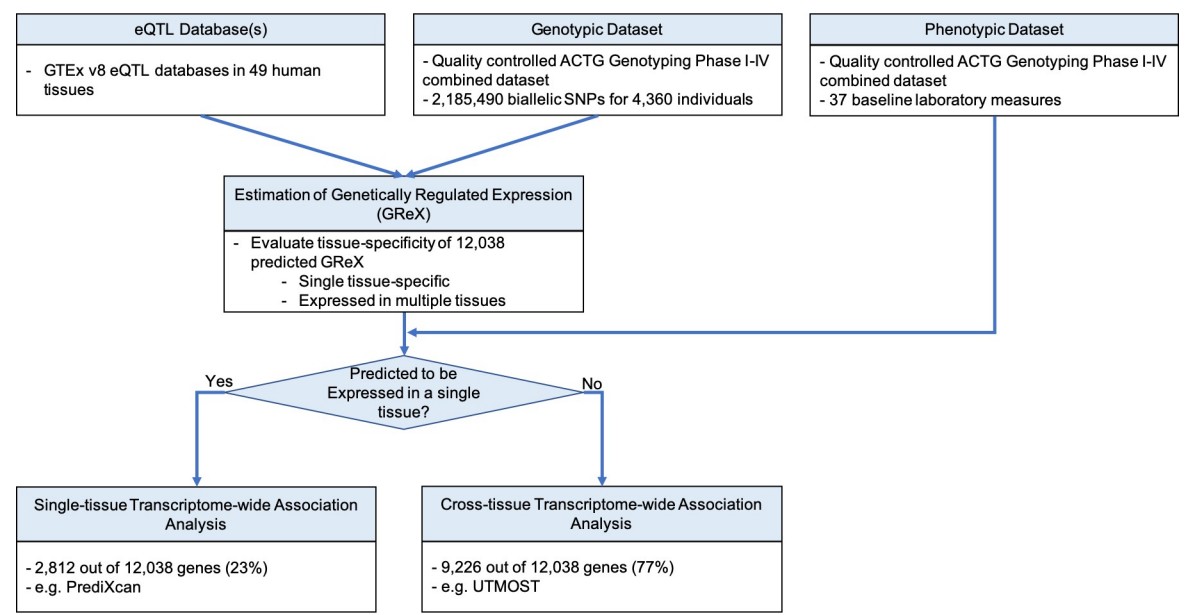

**Fig 7. The TSA-TWAS analytic framework for the ACTG combined genotyping phase I-IV baseline laboratory traits.** Approximately 2.2 million SNPs, 4,360 individuals, and 37 baseline laboratory traits survived the QC. UTMOST eQTL models were used to impute GReX of a total of 12,038 genes in 49 tissues. 2,812 genes (23%) had GReX in one single tissue, and 9,226 genes (77%) had GReX in more than one tissue.

a glycosylphosphatidylinositol-degrading enzyme that releases attached proteins from the plasma membrane and engages in regulation of alkaline phosphate activities. Other replicated discoveries included association between *ALDH5A1* and plasma alkaline phosphatase levels (p = 1.79×10⁻¹¹) [32], *C6orf48* and absolute basophil count (p = 1.69×10⁻¹²) [33], *KCNJ15* and plasma triglyceride levels (p = 3.18×10⁻¹³) [34].

We have additionally replicated several genes' association with plasma viral loads in HIV-positive adults, including *A4GALT* (p = 8.39×10⁻¹¹) [35], *ABCB4* (p = 1.07×10⁻¹¹) [36], *C4B* (p = 4.11×10⁻¹⁵) [37], *GABBR1*(p = 1.14×10⁻¹²) [38], and HLA-B (p = 1.15×10⁻¹¹) [39].

Fine-mapping of potential baseline laboratory measure-related genes retrieved a proof-of-concept association—*SORT1* association with plasma low density lipoprotein-cholesterol levels [40–42] (LDL-c; marginal posterior inclusion probability = 0.683, S3 Table).

## Novel genes prioritized by the TSA-TWAS

In addition to the above replications, TSA-TWAS identified novel associations with plasma viral load (Table 4). For instance, *PRDX5* (p = 7.01×10⁻¹⁴, which encodes a member of the peroxiredoxin family of antioxidant enzymes) was associated with plasma viral load with great significance. Several novel genes were first time reported to be associated with certain baseline laboratory values, which were otherwise associated with other traits by previous studies. For instance, *ATF6B* is a protein-coding gene that encodes a transcription factor in the unfolded protein response (UPR) pathway during ER stress and it has been associated with HIV-associated neurocognitive disorders in previous research [43]. In our study, ATF6B associates with plasma viral load (p = 2.83×10⁻⁹).

Several novel associations were further supported by colocalization analyses, for example, the association between *NLRC5* and fasting HDL (locus RCP = 0.0292 in adrenal gland).

**Table 2. Summary statistics of the ACTG genotyping phase I-IV baseline laboratories.**

| Trait | Sample Size | Mean | Std. Dev. | Min | Max | Transformation | Unit | Description |
|---|---|---|---|---|---|---|---|---|
| Albumin | 1216 | 4.05 | 0.44 | 1.80 | 5.30 | | g/dL | week 0 albumin (Alb, g/dL) |
| Bicarbonate | 3971 | 26.01 | 2.94 | 12.00 | 35.00 | | mmol/L | week 0 bicarbonate (Bicarb, mmol/L) |
| Calcium | 1336 | 9.17 | 0.44 | 7.40 | 10.80 | | mg/dL | week 0 calcium (Ca, mg/dL) |
| Chloride | 4048 | 103.27 | 2.94 | 88.00 | 117.00 | | mmol/L | week 0 chloride (Cl, mmol/L) |
| Cholesterol | 4286 | 159.27 | 36.80 | 5.90 | 414.00 | | mg/dL | week 0 cholesterol (Chol, mg/dL) |
| Creatinine | 4100 | 0.91 | 0.20 | 0.05 | 2.80 | | mg/dL | week 0 creatinine (Creat, mg/dL) |
| HDL-c | 2376 | 37.31 | 12.78 | 3.90 | 148.00 | | mg/dL | week 0 HDL-c (HDL-c, mg/dL) |
| Hemoglobin | 4293 | 13.49 | 1.77 | 6.00 | 20.20 | | g/dL | week 0 hemoglobin (Hgb, g/dL) |
| Absolute basophil count | 2526 | 1.44 | 0.32 | 0.00 | 3.39 | $Log_{10}$ | cells/mm3 | $log_{10}$ transformed week 0 absolute basophil count (Baso, cells/mm3) |
| Absolute eosinophil count | 3932 | 2.06 | 0.40 | 0.18 | 3.55 | $Log_{10}$ | cells/mm3 | $log_{10}$ transformed week 0 absolute eosinophil count (Eos, cells/mm3) |
| Alkaline phosphatase | 4226 | 1.88 | 0.15 | 0.70 | 2.72 | $Log_{10}$ | U/L | $log_{10}$ transformed week 0 alkaline phosphatase (AlkP, U/L) |
| ALT | 4233 | 1.48 | 0.27 | 0.04 | 2.81 | $Log_{10}$ | U/L | $log_{10}$ transformed week 0 ALT (ALT, U/L) |
| Absolute lymphocyte count | 4149 | 3.11 | 0.24 | 0.92 | 4.03 | $Log_{10}$ | cells/mm3 | $log_{10}$ transformed week 0 absolute lymphocyte count (Lymph, cells/mm3) |
| Absolute monocyte count | 4116 | 2.58 | 0.21 | 0.66 | 3.69 | $Log_{10}$ | cells/mm3 | $log_{10}$ transformed week 0 absolute monocyte count (Mono, cells/mm3) |
| Amylase | 1026 | 1.85 | 0.20 | 1.11 | 2.89 | $Log_{10}$ | U/L | $log_{10}$ transformed week 0 amylase (Amyl, U/L) |
| Absolute neutrophil count | 4277 | 3.32 | 0.21 | 2.28 | 4.67 | $Log_{10}$ | cells/mm3 | $log_{10}$ transformed week 0 absolute neutrophil count (ANC, cells/mm3) |
| AST | 4235 | 1.49 | 0.21 | 0.48 | 2.81 | $Log_{10}$ | U/L | $log_{10}$ transformed week 0 AST (AST, U/L) |
| BUN | 4221 | 1.08 | 0.15 | -0.22 | 2.17 | $Log_{10}$ | mg/dL | $log_{10}$ transformed week 0 BUN (BUN, mg/dL) |
| CK | 1360 | 1.97 | 0.38 | -0.05 | 3.79 | $Log_{10}$ | U/L | $log_{10}$ transformed week 0 CK (CK, U/L) |
| Fasting glucose | 3233 | 1.93 | 0.08 | 1.52 | 2.64 | $Log_{10}$ | mg/dL | $log_{10}$ transformed week 0 fasting glucose (Gluc fasting, mg/dL) |
| Glucose ($Log_{10}$) | 3031 | 1.93 | 0.08 | 1.70 | 2.77 | $Log_{10}$ | mg/dL | $log_{10}$ transformed week 0 glucose (Gluc, mg/dL) |
| LDL-c | 3539 | 1.95 | 0.16 | 0.00 | 2.57 | $Log_{10}$ | mg/dL | $log_{10}$ transformed week 0 LDL-c (LDL-c, mg/dL) |
| Lipoprotein | 1118 | 1.58 | 0.32 | 0.30 | 2.85 | $Log_{10}$ | | $log_{10}$ transformed week 0 lipoprotein |
| Platelet count | 4263 | 2.30 | 0.15 | 1.15 | 3.34 | $Log_{10}$ | x10E9/L | $log_{10}$ transformed week 0 platelet count (Plat, x10E9/L) |
| Total bilirubin | 4202 | -0.31 | 0.21 | -1.00 | 0.49 | $Log_{10}$ | mg/dL | $log_{10}$ transformed week 0 total bilirubin (TBili, mg/dL) |
| Triglyceride | 4318 | 2.07 | 0.25 | 1.08 | 3.45 | $Log_{10}$ | mg/dL | $log_{10}$ transformed week 0 triglyceride (Trig, mg/dL) |
| White blood cell count | 4279 | 0.62 | 0.16 | -0.05 | 1.49 | $Log_{10}$ | x10E3 cells/cu mm | $log_{10}$ transformed week 0 white blood cell count (WBC, x10E3 cells/cu mm) |
| Hematocrit | 4274 | 39.83 | 5.10 | 1.00 | 62.10 | | percent | week 0 hematocrit (Hct, percent) |
| Phosphate | 3261 | 3.44 | 0.61 | 0.80 | 7.70 | | mg/dL | week 0 phosphate (Phos, mg/dL) |
| Potassium | 4062 | 4.15 | 0.39 | 2.00 | 8.00 | | mmol/L | week 0 potassium (K, mmol/L) |
| Sodium | 4067 | 138.88 | 2.80 | 123.00 | 151.00 | | mmol/L | week 0 sodium (Na, mmol/L) |
| CD4 count | 4358 | 14.78 | 6.46 | 0.00 | 36.55 | Square root | cells/mm3 | square root of absolute CD4 count at week 0 |
| Viral load | 4358 | 4.75 | 0.72 | 2.02 | 7.11 | $Log_{10}$ | copies/dL | week 0 viral load RNA |
| Fasting cholesterol | 4136 | 158.42 | 36.24 | 6.10 | 414 | | mg/dL | week 0 fasting cholesterol |
| Fasting HDL-c | 4126 | 1.56 | 0.15 | 0.60 | 2.20 | $Log_{10}$ | mg/dL | $log_{10}$ transformed week 0 fasting HDL-c |
| Fasting LDL-c | 4042 | 1.95 | 0.15 | 0.85 | 2.57 | $Log_{10}$ | mg/dL | $log_{10}$ transformed week 0 fasting LDL-c |
| Fasting triglyceride | 3888 | 2.05 | 0.24 | 1.08 | 2.45 | $Log_{10}$ | mg/dL | $log_{10}$ transformed week 0 fasting triglycerides |

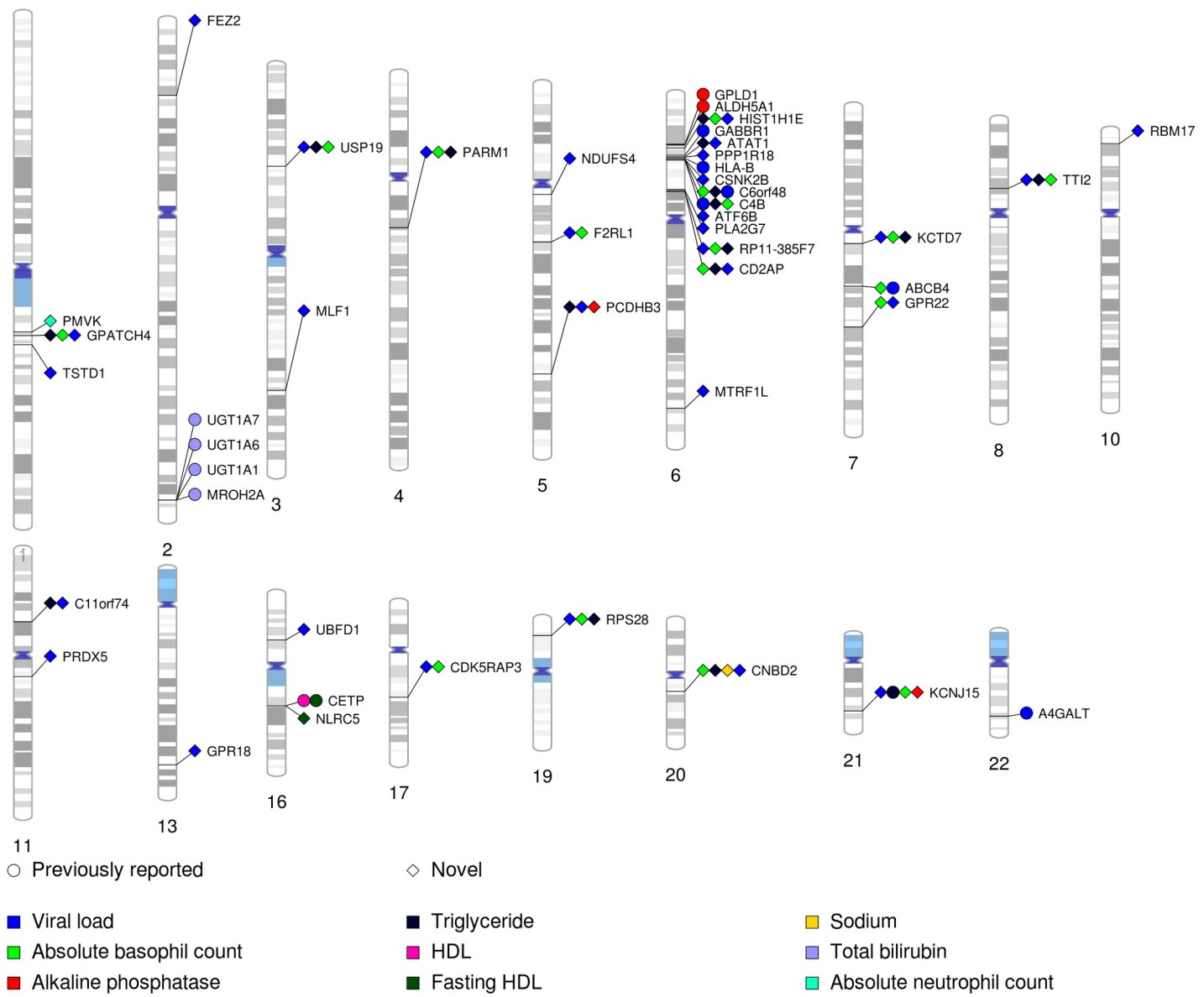

**Fig 8. PhenoGram of statistically significant gene-trait associations identified by the TSA-TWAS analytic framework.** We plotted the associations with p-value $< 1.12\times10^{-7}$. Each association is arranged according to the SNP location on each chromosome and the points are color-coded by baseline laboratory values. Diamonds represented previously reported or replicated associations, and circle represented novel associations identified in this study.

## Pleiotropic genes associated with baseline laboratory values

We also found several pleiotropic genes which were statistically significantly associated with plasma viral load, triglyceride levels, and/or absolute basophil count (Fig 8). These included *ABCB4*, *ATAT1*, *C11orf74*, *C4B*, *C6orf48*, *CD2AP*, *CDK5RAP3*, *CNBD2*, *F2RL1*, *GPATCH4*, *GPR22*, *KCNJ15*, *KCTD7*, *PARM1*, *PCDHB3*, *RPS28*, *TTI2*, *USP19*. Some of them were located on chromosome 6, surrounding the major histocompatibility complex (MHC) region, while the rest scattered across the human genome. Meanwhile, we did not observe correlations among plasma viral load, triglyceride levels, or absolute basophil count. The strongest

**Table 3. Replicated associations related to HIV baseline laboratory values identified by TSA-TWAS.**

| Trait | Gene | Chromosome | TSS | P | Colocalized Tissues | Locus RCP |
|---|---|---|---|---|---|---|
| Alkaline phosphatase | GPLD1 | 6 | 24428177 | 1.08E-11 | Esophagus | 0.1006 |
| | ALDH5A1 | 6 | 24494852 | 1.79E-11 | Liver | 0.1805 |
| Fasting HDL | CETP | 16 | 56961850 | 4.49E-12 | Adipose | 0.0916 |
| HDL | CETP | 16 | 56961850 | 4.49E-12 | Artery | 0.2837 |
| Total bilirubin | UGT1A6 | 2 | 233692866 | 2.78E-15 | | |
| | MROH2A | 2 | 233775679 | 1.39E-12 | | |
| | UGT1A1 | 2 | 233760248 | 3.59E-12 | Liver | 0.1318 |
| | UGT1A7 | 2 | 233681938 | 4.51E-12 | | |
| Triglyceride | KCNJ15 | 21 | 38256698 | 3.18E-13 | | |
| Viral load | C4B | 6 | 32014762 | 4.11E-15 | | |
| | GABBR1 | 6 | 29602228 | 1.14E-12 | | |
| | ABCB4 | 7 | 87401697 | 1.07E-11 | | |
| | HLA-B | 6 | 31269491 | 1.15E-11 | | |
| | C6orf48 | 6 | 31834608 | 2.32E-11 | | |
| | A4GALT | 22 | 42692121 | 8.39E-11 | | |

correlation was observed between plasma viral load and triglyceride levels ($r^2$ = 0.24), suggesting only weak correlation, and correlations for the other pairs of laboratory values were approximately 0. Overall, there were potential pleiotropic genes for plasma viral load, triglyceride levels, and/or absolute basophil count in HIV-positive adults.

**Table 4. Novel associations related to HIV baseline laboratory values identified by TSA-TWAS.**

| Trait | Gene | Chromosome | TSS | P | Colocalized Tissues | Locus RCP |
|---|---|---|---|---|---|---|
| Absolute basophil count | KCTD7 | 7 | 66628767 | 3.08E-14 | | |
| | CNBD2 | 20 | 35955360 | 3.83E-13 | | |
| | CD2AP | 6 | 47477789 | 7.27E-13 | | |
| | RP11-385F7.1 | 6 | 47477243 | 1.32E-12 | | |
| | C6orf48 | 6 | 31834608 | 1.69E-12 | | |
| | PARM1 | 4 | 74933095 | 1.84E-11 | | |
| | USP19 | 3 | 49108046 | 1.51E-10 | | |
| | GPATCH4 | 1 | 156594487 | 2.24E-10 | | |
| | GPR22 | 7 | 107470018 | 1.81E-09 | | |
| | HIST1H1E | 6 | 26156354 | 2.19E-09 | | |
| | RPS28 | 19 | 8321500 | 2.87E-09 | | |
| | KCNJ15 | 21 | 38256698 | 4.72E-09 | | |
| | TTI2 | 8 | 33473423 | 6.35E-09 | | |
| | CDK5RAP3 | 17 | 47967810 | 1.05E-08 | | |
| | F2RL1 | 5 | 76818933 | 2.99E-08 | | |
| | C4B | 6 | 32014762 | 8.92E-08 | | |
| Absolute neutrophil count | PMVK | 1 | 154924734 | 3.63E-08 | Adipose | 0.0663 |
| Alkaline phosphatase | PCDHB3 | 5 | 141100756 | 7.44E-09 | Artery | 0.1294 |
| | KCNJ15 | 21 | 38256698 | 2.77E-08 | | |
| Fasting HDL | NLRC5 | 16 | 56989485 | 1.70E-09 | Adrenal Gland | 0.0292 |
| Sodium | CNBD2 | 20 | 35955360 | 7.71E-08 | | |

*(Continued)*

**Table 4.** (Continued)

| Trait | Gene | Chromosome | TSS | P | Colocalized Tissues | Locus RCP |
|---|---|---|---|---|---|---|
| Triglyceride | PCDHB3 | 5 | 141100756 | 5.78E-14 | | |
| | GPATCH4 | 1 | 156594487 | 2.12E-12 | | |
| | CNBD2 | 20 | 35955360 | 7.21E-12 | | |
| | C6orf48 | 6 | 31834608 | 9.13E-12 | | |
| | PARM1 | 4 | 74933095 | 1.88E-11 | | |
| | TTI2 | 8 | 33473423 | 2.69E-11 | | |
| | USP19 | 3 | 49108046 | 9.69E-11 | | |
| | HIST1H1E | 6 | 26156354 | 1.20E-10 | | |
| | CD2AP | 6 | 47477789 | 1.04E-09 | | |
| | RP11-385F7.1 | 6 | 47477243 | 1.17E-09 | | |
| | C4B | 6 | 32014762 | 1.23E-09 | | |
| | KCTD7 | 7 | 66628767 | 1.34E-08 | | |
| | RPS28 | 19 | 8321500 | 1.40E-08 | | |
| | C11orf74 | 11 | 36594493 | 1.94E-08 | | |
| | ATAT1 | 6 | 30626842 | 5.32E-08 | | |
| Viral load | PPP1R18 | 6 | 30676389 | 6.27E-14 | | |
| | PRDX5 | 11 | 64318088 | 7.01E-14 | | |
| | F2RL1 | 5 | 76818933 | 1.81E-12 | | |
| | CDK5RAP3 | 17 | 47967810 | 1.95E-12 | | |
| | RPS28 | 19 | 8321500 | 3.50E-12 | | |
| | USP19 | 3 | 49108046 | 3.60E-12 | | |
| | KCTD7 | 7 | 66628767 | 3.84E-12 | | |
| | TTI2 | 8 | 33473423 | 4.27E-12 | | |
| | TSTD1 | 1 | 161037631 | 4.57E-12 | | |
| | UBFD1 | 16 | 23557732 | 5.27E-12 | | |
| | RP11-385F7.1 | 6 | 47477243 | 1.05E-11 | | |
| | KCNJ15 | 21 | 38256698 | 1.08E-11 | | |
| | CD2AP | 6 | 47477789 | 1.54E-11 | | |
| | CNBD2 | 20 | 35955360 | 1.70E-11 | | |
| | PARM1 | 4 | 74933095 | 1.86E-11 | | |
| | ATAT1 | 6 | 30626842 | 2.20E-11 | | |
| | HIST1H1E | 6 | 26156354 | 8.44E-11 | | |
| | MTRF1L | 6 | 152987362 | 1.14E-10 | | |
| | MLF1 | 3 | 158571163 | 1.23E-10 | | |
| | PCDHB3 | 5 | 141100756 | 2.42E-09 | | |
| | ATF6B | 6 | 32115335 | 2.83E-09 | | |
| | GPR22 | 7 | 107470018 | 3.75E-09 | | |
| | RBM17 | 10 | 6088987 | 5.39E-09 | | |
| | PLA2G7 | 6 | 46704320 | 6.34E-09 | | |
| | GPATCH4 | 1 | 156594487 | 1.81E-08 | | |
| | NDUFS4 | 5 | 53560633 | 2.07E-08 | | |
| | C11orf74 | 11 | 36594493 | 2.14E-08 | | |
| | CSNK2B | 6 | 31665391 | 2.76E-08 | | |
| | GPR18 | 13 | 99254714 | 4.10E-08 | | |
| | FEZ2 | 2 | 36531805 | 4.54E-08 | | |

## Discussion

### Novel design of the simulation framework

In this report, we described a novel simulation framework for TWAS, and evaluated TWAS gene prioritization performance for genes with various degrees of tissue specificity. Our simulation results validated conclusions from several previous eQTL or TWAS studies [13–15,21], and also generated new findings that warrant attention in future TWAS. First, TWAS methods tested in this study all had well-controlled type I error rates ($\leq$ 5%) for genes with any degrees of tissue-specificity. Second, single-tissue TWAS tended to have higher false positive rates of tissues. The phenomenon became more obvious when genes had more correlated expression levels across tissues. For tissue-specific genes, false positive rates of tissues could be controlled ($\leq$ 5%) by adopting a more stringent multiple testing correction approach. However, for ubiquitously expressed genes, false positive rates of tissues remained significant (~77%) even after a stringent multiple testing adjustment. Third, TWAS gene prioritization power was improved by eQTLs that were identified by jointly analyzing transcriptomic data across tissues. Fourth, for tissue-specific genes, single-tissue and cross-tissue gene-level association approaches had similar power. For ubiquitously expressed and similarly expressed genes, cross-tissue association approaches had greater power.

We further tested our simulation designs for how they would translate to real-world data by evaluating trait heritability in our simulated datasets. We found no apparent effect of MAF distribution on trait heritability under TWAS models. Instead, trait heritability increased with $R^2_{expression-trait}$. When $R^2_{expression-trait} = 1\%$, trait heritability was approximately 1% (s.e. = 0.059%). The estimated trait heritability was within a reasonable range and supported our simulation design.

### Associations in the clinical trials dataset

TSA-TWAS successfully replicated proof-of-concept gene-trait associations, including associations between *CETP* and HDL-c, and between *GPLD1* and plasma alkaline phosphatase levels. For total plasma bilirubin levels, our TSA-TWAS framework prioritized *UGT1A1* and genes near *UGT1A1*. These genes span 1Mbp at the 2q37.1 locus and are within the same topologically associating domain (TAD), which suggests that a regulatory mechanism may affect expression of the entire *KCNJ13-UGT1A-MROH2A* gene region. Multiple associations at a risk locus suggested possible transcriptional regulation that targets the whole genetic region. However, shared transcriptional regulation of neighboring genes does not indicate the same phenotypic impact. While many genes in the 2q37.1 locus have been associated with total bilirubin levels in numerous studies [25–28], *UGT1A1* is the only known functional gene that encodes the hepatic protein to glucuronidates bilirubin in liver [29]. This discovery indicated that TWAS was likely to assign statistical significance to neighboring genes as a result of shared transcriptional regulation or LD structure [18]. Understanding of complex trait regulatory mechanisms is difficult to achieve with GWAS and gene expression data alone. Functional genomics data, computational methods, and validation experiments are required to identify causal genes and mechanisms for a risk locus.

TSA-TWAS has also identified several pleiotropic genes that linked plasma viral load, absolute basophil count, and/or triglyceride levels, which were otherwise independent from each other. Plasma viral load is a strong predictor of clinical outcome and is highly variable among people living with HIV. Individuals vary in their ability in suppressing viral loads, in the absence of antiretroviral treatments. Moreover, people living with HIV experience dyslipidemia to different degrees at baseline or after antiretroviral therapy (ART) and, thus, have higher risk of developing cardiovascular diseases than those living without HIV. HIV-associated and ART-induced dyslipidemia imposes challenges in clinical care of comorbid cardiovascular diseases

risks for people living with HIV [44]. The discovery of pleiotropic genes demonstrates the complexity of gene expression and genetic architecture of HIV baseline lab values. The complicated inter-individual variability across multiple traits may be of interest for future research exploring HIV pathogenesis and treatment responses. We also acknowledged that some of the pleiotropic genes are located in the MHC region, which has a complicated LD structure. We defer this question to future research with deep-genotyping or sequencing of specific *HLA* regions.

## Limitations & future directions

Our simulations revealed high false positive rates of tissues for single-tissue TWAS. The high false positive rates seen with single-tissue TWAS may be due to limited sample sizes for eQTL discovery. GTEx analysis has shown that discovery of tissue-specific eQTLs is contingent on the sample sizes of tissues [10]. Unfortunately, many tissues still have limited sample sizes for the identification of tissue-specific eQTLs. Consequently, single-tissue TWAS may not have ample power to prioritize potential trait-related tissues. Adopting stricter multiple testing adjustment strategies for single-tissue TWAS is one practical approach to help reduce false positive rates in prioritized tissues, but this will sacrifice power.

The evaluation of TWAS power and type I error rates estimated from this simulation study might be limited due to the small sample sizes (N = 2,000 for association analyses). We selected this sample size for simulation in order to make it comparable to the average sample size of the ACTG phase I-IV combined clinical traits interrogated in this study. TWAS gene prioritization power can be improved with greater sample, but also under influence of many other factors as shown in Veturi *et al.* [21] and this study. Thus, TWAS performance can differ from dataset to dataset when using different TWAS methods. It was difficult to take every factor into consideration in this work. We dedicated this study to explore tissue specificity's impact on TWAS performance, and, for future TWAS studies, suggest customized simulation to better understand TWAS performance on specific datasets and diseases of interest.

Pinpointing complex trait-related genes remains a challenge that is beyond the scope of our study here. The causes of this challenge are multi-faceted, including co-expression of neighboring genes [45]), correlated SNPs or eQTLs at a locus (i.e. LD), bias and noises from trait-irrelevant tissues [18], etc. Some of the issues were comprehensively described and discussed in [18]. Our TSA-TWAS aimed at improving power to identify associated genes. For the purpose of identifying trait-related genes, TSA-TWAS should be followed by a fine-mapping analysis (FOCUS [24]). FOCUS identifies credible sets of potential trait-related genes by addressing the issues of LD and co-regulation of genes in TWAS. In our ACTG TSA-TWAS analysis, we further performed colocalization analysis using fastENLOC [46] for all genes in the FOCUS identified credible set to prioritize genes for future research. Some associations, for example, *UGT1A1*-total bilirubin levels (locus RCP = 0.1318 in liver), were supported by the colocalization results. However, many statistically significant associations were not supported by colocalization likely due to the conservative nature of colocalization [47].

## Conclusions

Gene-level association studies offer the opportunity to better understand the genetic architecture of complex human traits by leveraging regulatory information from both noncoding and coding regions of the genome. This may expedite translation of basic research discoveries to clinical applications. We provide a comprehensive simulation algorithm to fully investigate TWAS performance for diverse biological scenarios. Based on our simulation, we promote a TSA-TWAS analytic framework. TSA-TWAS framework on ACTG clinical trials data ascribed statistical significance to proof-

of-concept gene-trait associations, and also found several novel associations and pleiotropic genes, suggesting the complexity of HIV-related traits that latest bioinformatics methods can reveal.

Additional work is needed to fully understand the tissue and genetic architecture underlying complex traits. The simulation algorithm and schema developed for this study is versatile enough to answer other questions regarding causal genes and tissues for complex traits. Overall, our work provides and tests a novel, flexible simulation framework and an TSA-TWAS analytic framework for future complex trait studies.

# Materials and methods

## TWAS simulation design

The simulation study systematically evaluated how the tissue-specificity of eQTLs and gene expression levels influences TWAS gene prioritization performance. We assumed additive genetic effects of eQTLs on gene expression levels, and of gene expression levels on traits. The TWAS simulation scripts are available in R programming language at GitHub (https://github.com/RitchieLab/multi_tissue_twas_sim).

**Genotype.** We started by simulating genotypes for one gene in 1,500 individuals, which include eQTL and non-eQTL SNPs. Genotypes are denoted as $X_{N \times M}$ throughout this paper, where $N$ denotes the total number of individuals and $M$ denotes the total number of SNPs in a gene that include tissue-specific eQTLs, multi-tissue eQTLs and non-eQTL SNPs. These individuals were later stratified into an eQTL discovery dataset ($N_{eQTL}$ = 500) and a TWAS testing dataset ($N_{TWAS}$ = 1000), sample sizes comparable to those of current GTEx and ACTG datasets used in this analysis, respectively. Genotypes were simulated as biallelic SNPs and then converted into allele dosages as is done in most eQTL detection methods. MAF assigned to SNPs raged from 1% to 50% and were randomly drawn from a uniform distribution, $U$(0.01, 0.5). Parameter settings of eQTLs in this simulation were drawn from observations in different eQTL databases (S1–S3 Figs).

**Gene expression level.** We simulated one gene's standardized expression levels at a time such that it was expressed in a fixed number of tissues. Let $P$ denote the number of tissues where the gene is expressed, $P$ = 1, 2, 5, or 10. If a gene is only expressed in a single tissue ($P$ = 1), then, only single-tissue eQTLs were simulated for this given gene and no multi-tissue eQTLs were present.

A previous study showed that the number of eQTLs in a gene does not have as pronounced an effect on the TWAS power in comparison to other parameters [21]. Hence, we assumed that a given gene was regulated by the same total number of eQTLs in each of the $P$ tissues, which is denoted by $M_{eQTLs}$ ($M_{eQTLs}$ = 30). eQTLs can be tissue-specific or have effect across multiple tissues. Here, we defined tissue-specific eQTLs as those that had effects in one and only one tissue. Multi-tissue eQTLs were defined as those who had effects in all $P$ tissues in which the given gene is simulated to be expressed. We allowed multi-tissue eQTLs to have different effect sizes in different tissues. Assuming that a gene was expressed in $P$ tissue(s) (say $P$ = 5), then, this gene is regulated by both, tissue-specific eQTLs and multi-tissue eQTLs, in any of the $P$ tissues. Let $M_{ts-eQTLs}$ denote the number of tissue-specific eQTLs, and $M_{mt-eQTLs}$ the number of multi-tissue eQTLs. A simulated gene had the same $M_{ts-eQTLs}$ across $P$ tissues, and the same $M_{mt-eQTLs}$ across $P$ tissues, such that $M_{ts-eQTLs}$ and $M_{mt-eQTLs}$ added up to $M_{eQTLs}$ in each of the $P$ tissues. Five different numbers of $M_{mt-eQTLs}$ (0, 6, 12, 18, 24, corresponding $M_{st-eQTLs}$ = 30, 24, 18, 12, 6) were evaluated, except when a gene was simulated to be expressed only in one gene, in which case $M_{mt-eQTLs}$ always equaled 0.

Each gene was simulated under an additive genetic model per tissue. Let $E_{N \times P}$ denote the simulated gene expression levels for one gene, of $N$ individuals, and across $P$ tissues. For the

given simulated gene, let $E_{np}$ represent the simulated expression level of the $n$th individual in the $p$th tissue, which is an aggregate of tissue-specific eQTLs, multi-tissue eQTLs and non-eQTL effects in individual $n$ for tissue $p$. The multivariate normal random effects model to simulate one gene's expression levels is then expressed as follows:

$$E = X_{ts-eQTLs}\beta_{ts-eQTLs} + X_{mt-eQTLs}\beta_{mt-eQTLs} + \varepsilon_1$$

where $E$ is the $N{\times}P$ matrix of standardized gene expression levels for a gene in $N$ individuals across $P$ tissues. $X_{ts-eQTLs}$ is the $N{\times}M_{ts-eQTLs}$ matrix of standardized tissue-specific eQTL genotypes. Similarly, $X_{mt-eQTLs}$ is the $N{\times}M_{mt-eQTLs}$ matrix of standardized multi-tissue eQTL genotypes. $\beta_{ts-eQTLs}$ is a $M_{ts-eQTLs}{\times}P$ matrix of tissue-specific eQTL effects. $\beta_{ts-eQTLs,ip}$ represents the $i$th tissue-specific eQTL in the $p$th tissue, which could be a different eQTL across $P$ tissues. Each value in the $\beta_{ts-eQTLs}$ is independent of the others. $\beta_{mt-eQTLs}$ is a $M_{mt-eQTLs}{\times}P$ matrix of multi-tissue eQTL effects wherein $\beta_{mt-eQTLs,jp}$ represents the $j$th multi-tissue eQTL in the $p$th tissue. In contrast to tissue-specific eQTLs, $\beta_{mt-eQTLs,j.}$ denotes the same $j$th multi-tissue eQTL in all $P$ tissues, and is allowed to have similar or dissimilar effect sizes across $P$ tissues (explained later in this section). $vec(\beta_{ts-eQTLs}) \sim N(\mathbf{0}_{M_{ts-eQTLs}\times P}, \Sigma_{P\times P}^{ts-eQTLs} \otimes I_{M_{ts-eQTLs}})$

$$where\ \Sigma_{P\times P}^{ts-eQTLs} = \begin{cases} h_{SNP-expression}^2 \times \dfrac{M_{ts-eQTLs}}{M_{eQTLs}}, p = p' \\ 0, p \neq p' \end{cases}$$ . The constant, $h_{SNP-expression}^2 \times \frac{M_{ts-eQTLs}}{M_{eQTLs}}$,

represents the proportion of variation in gene expression that can be explained by tissue-specific eQTLs. $vec(\beta_{mt-eQTLs}) \sim N(\mathbf{0}_{M_{mt-eQTLs}\times P}, \Sigma_{P\times P}^{mt-eQTLs} \otimes I_{M_{mt-eQTLs}})$

$$where\ \Sigma_{P\times P}^{mt-eQTLs} = \begin{cases} h_{SNP-expression}^2 \times \dfrac{M_{mt-eQTLs}}{M_{eQTLs}}, p = p' \\ cor\left(tissue_p, tissue_{p'}\right) \times h_{SNP-expression}^2 \times \dfrac{M_{mt-eQTLs}}{M_{eQTLs}}, p \neq p' \end{cases}$$ . The constant,

$h_{SNP-expression}^2 \times \frac{M_{mt-eQTLs}}{M_{eQTLs}}$, represents the proportion of gene expression variation that can be explained by multi-tissue eQTLs. $cor(tissue_p, tissue_{p'})$ represents the extent of similarity between $\beta_{mt-eQTLs,.p}$ and $\beta_{mt-eQTLs,.p'}$, i.e. the Pearson Correlation Coefficient between multi-tissue eQTL effect sizes in the $p$th and $p'$th tissues, respectively. The simulation algorithm allows multi-tissue eQTLs to have five different levels of $cor(tissue_i, tissue_j)$ (0, 0.2, 0.4, 0.6, and 0.8). $\varepsilon_1$ is the $N{\times}P$ matrix of residual errors that represent non-eQTL effects on a gene's expression level and $vec(\varepsilon_1) \sim N(\mathbf{0}_{N\times P}, \Sigma_{P\times P}^e \otimes I_n)$

$$where\ \Sigma_{P\times P}^e = \begin{cases} 1 - h_{SNP-expression}^2, p = p' \\ cor(tissue_p, tissue_{p'}) \times 1 - h_{SNP-expression}^2, p \neq p' \end{cases}$$ . The constant,

$1 - h_{SNP-expression}^2$, represents the proportion of gene expression variation that can be explained by factors other than eQTLs that can also regulate a gene's final transcription isoforms and levels. We designed the error term to have such a covariance structure that the final aggregate expression levels of the given gene in $p$th tissue ($E_{.p}$) was correlated with that in the $p'$th tissue ($E_{.p'}$) due to multi-tissue eQTLs as well as other biological factors. These other biological factors (such as alternative splicing events, post-transcriptional modifications and regulation of mRNA degradation) can either be shared or different across tissues. We adopted a simple assumption that the more similar a gene's expression levels are across tissues, the more likely multi-tissue eQTLs (and non-eQTL biological factors) will share effect sizes across tissues. Thus, correlation of gene expression across tissues (for example, correlation between $E_{.p}$ and $E_{.p'}$) is expected to be similar to, if not the same as, the correlation of multi-tissue eQTL effect sizes (for example, correlation between $\beta_{mt-eQTLs,.p}$ and $\beta_{mt-eQTLs,.p'}$) as well as the correlation

between non-eQTL biological factors. All three random effect terms, i.e. $\beta_{ts-eQTLs}$, $\beta_{mt-eQTLs}$, and $\varepsilon_1$ were simulated using the *rmvnorm* function from the R package, *mvtnorm*. We evaluated the extent of bias between assumed combination of simulation parameters and those estimated from the empirical distribution of simulated $E_{N \times P}$, which met the expectation (S14 Fig).

In the special case where a gene was simulated to be expressed only in a single tissue, the model was equivalent to a univariate normal distribution with mean 0 and variance equal to the expression heritability of that gene.

Tissue specificity of genes was characterized by the number of tissues in which genes are expressed as well as the similarity of gene expression levels across tissues. Tissue specificity of eQTLs was characterized by the proportion of multi-tissue eQTLs in a gene, the number of tissues where multi-tissue eQTLs were effective, and the similarity of eQTL effect sizes across tissues.

**Phenotype.** We assumed one and only one causal tissue for a phenotype and simulated phenotype datasets for the TWAS testing dataset ($N = 1,000$). This design was adopted from the simulation work of Dr. Yiming Hu *et al.* in the paper that described UTMOST [15]. Let $E_{eQTLs}$ denote the standardized genetically regulated expression component in the causal tissue. The model to simulate traits from gene expression levels can be expressed as $Y = E_{eQTLs}b_1 + \varepsilon_2$, where $Y$ is a 1000×1 vector of standardized responses for the 1,000 individuals in the TWAS testing dataset, $b_1$ is the $M_{eQTLs} \times 1$ vector of gene expression effect drawn from a normal distribution with mean zero and variance $R^2_{expression-trait}$, and $\varepsilon_2$ is the vector of normally-distributed errors with mean zero and variance 1- $R^2_{expression-trait}$. $R^2_{expression-trait}$ was assigned values in 0.001%, 0.05%, 0.5% and 1%, to represent different strengths of gene expression level-trait relations. To evaluate type I error rates, $R^2_{expression-trait} = 0\%$ corresponded to the null model where gene and trait were unrelated.

**eQTL detection.** We adopted two types of eQTL detection methods, 1) elastic net (implemented in PrediXcan [5]) and 2) group LASSO (implemented in UTMOST [15]). For ease of parallel computation, these two algorithms were adapted and integrated into the TWAS simulation tool scripts. eQTLs detected in a single tissue context (elastic net) and those detected in an integrative tissue context (group LASSO) were then used to impute GReX, and for gene-trait association analyses.

**Imputation of GReX.** Expression level of a gene can be imputed using a linear model as $E = X\beta$, where $E$ is the $N \times 1$ vector of imputed gene expression levels of the gene, $X$ is the $N \times M$ matrix of genotypes, and $\beta$ is the $M \times 1$ vector of eQTLs' estimated regulatory effects on the gene, and can be obtained by either elastic net or group LASSO.

**Association analysis.** Single-tissue gene-trait associations were then estimated using SLR model, i.e., *lm* function in R. Cross-tissue gene-trait association analyses were also conducted in R but using PC Regression (implemented in MulTiXcan [16]) and GBJ test (implemented in UTMOST [15]).

**Measures of TWAS performance.** Each combination of simulation parameters was repeated 100 times independently to assess power and type I error rates at $\alpha = 0.05$. Estimation of TWAS power was calculated as the percentage that a simulated causal gene was successfully identified as statistically significant in the causal tissue in the hundred simulations. Estimation of TWAS type I error rates were calculated as the percentage that a gene was falsely identified as statistically significant when there was no gene-trait signal simulated in the hundred simulations. We assumed that a gene is related to a trait in a single tissue, which is often the case for non-pleiotropic genes. In the simulation, we knew the causal tissues for the simulated traits. We calculated the false positive rates of tissues by counting the proportion of statistically significant results that were in non-causal tissues.

The entire process was repeated 20 times for each combination of simulation parameters to avoid sampling variability and to determine distributions of power, type I error rates, and false positive rates of tissues. We further evaluated the statistical significance of the differences in power and type I error rate between every pairs of TWAS methods using Wilcoxon Signed-rank test.

## Evaluation of simulated genetic scenarios

Trait heritability assessment validated and supported our design of simulation parameters. We designed a Monte Carlo simulation approach to randomly generate eQTL-gene-trait relations using the aforementioned simulation tool. Each replication simulated one genotypic dataset and one subsequent GReX profile for a gene. We simulated 30 non-eQTL and 30 eQTL SNPs for 5,000 individuals in which MAF followed a uniform distribution of 1–50% and eQTLs explained 30% of gene expression variation. The GReX profile was then used to generate 50 different traits using different random seeds. Thus, each simulated genotypic dataset had 50 estimated trait heritability values available; we took the average of these as the point estimate of the trait $h^2$ for each genotypic dataset. GCTA [48] was not appropriate for our simulation as it assumes genome-wide genotypic data. Instead, we used the R package, *regress*, to estimate trait heritability in the simulated datasets. The entire process was repeated 30 times to generate a distribution of estimated trait heritability for a given combination of simulation parameters.

To determine the influence of MAF on trait heritability, we designed different ranges of MAF distributions. MAF of SNPs followed a uniform distribution of 1–50% as in the primary TWAS performance evaluation, and also 1–20% and 1–5%. We also simulated traits where $R^2_{expression-trait} = 0\%$ (negative control), 2%, or 5% (positive controls) to support the estimation of trait heritability when $R^2_{expression-trait} = 0.001\%$, 0.05%, 0.5%, and 1%.

## AIDS Clinical Trials Group studies

The ACTG is the world's largest HIV clinical trials network. It has conducted major clinical trials and translational biomedical research that have improved treatments and standards of care for people living with HIV in the United States and worldwide. In this study, we used data from four separate genotyping phases of specimens from ACTG studies in a combined dataset that comprises HIV treatment-naïve participants at least 18 years of age enrolled in randomized treatment trials [49–55]. Participants enrolled into ACTG protocols A5095, A5142, ACTG 384, A5202 or A5257. Informed consent for genetic research was obtained under ACTG protocol A5128. Clinical trial designs and outcomes, and results of a genome-wide pleiotropic study for baseline laboratory values have been described elsewhere [25,26].

## Genotypic data and quality control

A total of 4,411 individuals were genotyped in four phases. Phase I (samples from study A5095) was genotyped using Illumina 650Y array; Phase II (studies ACTG384 and A5142) and III (study A5202) were genotyped using Illumina 1M duo array; Phase IV (study 5257) was genotyped using Illumina HumanCoreExome BeadChip. Preparation of genotypic data included pre-imputation quality control (QC), imputation, and post-imputation QC. Pre- and post-imputation QC followed the same guidelines [56] and used PLINK1.90 [57] and R programming language. Imputation was performed on the combined ACTG phase I-IV genotype dataset after pre-imputation QC, which used IMPUTE2 [58] with 1000 Genomes Phase 1 v3 [59] as the reference panel. Combined ACTG phase I-IV imputed data comprised 27,438,241 variants. The following procedures/parameters were used in the post-imputation QC by PLINK1.90: sample inclusion in the ACTG genotyping phase I-IV phenotype collection, biallelic SNP check, imputation score ($> 0.7$),

concordance of genetic and self-reported sex, genotype call rate ($> 99\%$), sample call rate ($> 98\%$), MAF ($> 5\%$), and relatedness check ($\hat{\pi} > 0.25$; one individual was dropped from each related pair). Subsequent principal component analysis (EIGENSOFT [60]) projected remaining individuals onto the 1000 Genomes Project Phase 3 sample space to examine for population stratification. Based on percent of variance explained, the first three principal components estimated by SmartPCA in EIGENSOFT were used as covariates to adjust for population structure in the subsequent analyses. The final QC'ed ACTG phase I-IV combined imputed data comprised 2,185,490 genotyped and imputed biallelic SNPs for 4,360 individuals.

## Phenotypic data and QC

Data for 37 baseline (i.e., pre-treatment) laboratory measures were available from 5,185 HIV treatment-naive individuals in the ACTG genotyping phase I-IV datasets. We assembled these laboratory traits using a MySQL database and applied QC using R. We retained only individuals with available genotype data, and traits that were normally distributed and met the criterion of phenotype missing rate $< 80\%$. Frequency distributions of traits were inspected using *hist_-plot.R* that facilitates manual inspection of continuous traits by providing fast, high-throughput visualization along with necessary summary statistics of each visualized traits[61]. *hist_plot.R* is part of the CLARITE [61], which is available at https://github.com/HallLab/clarite. We also cross-referenced the retained traits to other published work that analyzed the same traits using these clinical trials datasets [25,26]. Non-fasting serum lipid measures were retained based on data from several studies [62–64]. The final combined dataset for ACTG genotyping phases I-IV comprised 37 baseline laboratory traits (Table 2).

## Description of a general TSA-TWAS analytic framework

The TSA-TWAS analytic framework has the following general steps.

1. Impute the GReX for the gene based on the input eQTL database(s) and the genotypic dataset.

2. Determine whether the gene is predicted to be expressed in only one tissue or in multiple tissues.

3. If the gene is predicted to be expressed in only one tissue, perform single-tissue TWAS using simple linear or logistic regression depending on the trait.

4. If the gene is predicted to be expressed in multiple tissues, perform cross-tissue TWAS using the GBJ test.

5. Repeat step 2–4 for the next gene.

6. (Optional) If there is more than one trait, repeat step 1–5 for the next trait.

## Imputation of GReX for genes

We used GTEx v8 MASHR-based eQTLs models [65] to impute gene expression levels in a tissue-specific manner. MASHR-based eQTLs models selected variants that have biological evidence of a potential causal role in gene expression, and estimated these variants' effect sizes on gene expression levels in 49 tissues, using GTEx v8 as the reference dataset (available at http://predictdb.org/). The GTEx v8 MASHR-based eQTLs models were downloaded from their website on October 31, 2019. The QC'ed ACTG phase I-IV combined imputed data was used to impute the individual-level GReX in 49 human tissues.

## Statistical analysis for gene-level associations

We tested for single-tissue gene-trait associations by performing association tests on imputed GReX and ACTG baseline lab traits using PLATO [66,67] in 49 tissues, separately. All baseline laboratory traits were continuous and thus were modeled by linear regression with covariates. Covariates included age, sex, and the first three principal components calculated by EIGENSOFT to adjust for sampling biases and underlying population structure. For cross-tissue association analyses, we adapted the UTMOST script in R programming language and performed the GBJ test for the individual-level ACTG data. The lowest p-value that can be generated by GBJ test in R is approximately $1 \times 10^{-15}$. No obvious inflation was observed in the TSA-TWAS framework. ACTG phenome-wide TWAS results were visualized using PhenoGram [68], a web-based, versatile data visualization tool to create chromosomal ideograms with customized annotations, available at http://visualization.ritchielab.org/phenograms/plot. Supplementary manhattan plot was created by *hudson*, a R package available at https://github.com/anastasia-lucas/hudson.

We further identified the credible set of baseline laboratory measure-associated genes to capture the full pool of possible causal genes. The analyses were done using the FOCUS [24] with the GWAS summary statistics of the 38 baseline laboratory measures and recommended multiple tissue, multiple reference panel eQTL databases that are provided on the FOCUS GitHub repository. LD information was computed from the quality-controlled genotype data of the ACTG participants.

We then performed colocalization analyses to see if any genes in the credible set colocalized with eQTL signals in any tissues. We first performed statistical fine-mapping of likely causal variants and calculated GWAS posterior inclusion probability (PIP) by applying a Bayesian method, TORUS [69], on the ACTG GWAS summary statistics. Then, we estimated the probability of colocalization between each of the GWAS and cis-eQTL signals using fastENLOC developed by Pividori et al. [46], following the guideline on the fastENLOC GitHub repository (https://github.com/xqwen/fastenloc). The locus RCP for each signal cluster of interest was calculated automatically by fastENLOC.

## Statistical correction

Two strategies to correct for multiple testing were implemented in the ACTG analysis, method-wise and family-wise Bonferroni significance thresholds. The method-wise approach ascribes significance to statistical tests by controlling for the number of tests conducted in one type of method. For single-tissue gene-trait associations, the method-wise Bonferroni significance threshold was corrected for the number of genes (n = 483) and traits (n = 37), which resulted in $\alpha = \frac{0.05}{2,812 \times 37} \approx 4.8 \times 10^{-7}$. For cross-tissue gene-trait associations, the method-wise Bonferroni significance threshold corrected for the number of genes and traits, which gave $\alpha = \frac{0.05}{9,226 \times 37} \approx 1.46 \times 10^{-7}$. The family-wise approach assigns significance to tests by accounting for all tests performed in this study to control for FWER. Hence, single-tissue and cross-tissue association tests shared the same family-wise Bonferroni significance threshold, $\frac{0.05}{12,038 \times 37} \approx 1.12 \times 10^{-7}$. The significance threshold for interpreting results, by default, referred to the family-wise threshold. All results reported are exact p-values and thus, can be easily compared to either multiple testing threshold.

## Supporting information

**S1 Fig. MAF distribution of GTEx v7 eQTLs.** MAF of eQTLs closely resembled a uniform distribution, ranging between 1% to 50%, with a spike around 20–25%.
(TIF)

**S2 Fig.** Distribution of number of eQTLs for a gene from (A) GTEx v7, (B) PredictDB eQTL datasets and (C) UTMOST eQTL datasets.
(TIF)

**S3 Fig. eQTL weight distribution for genes of different levels of tissue specificity.** (A) and (B) For tissue-specific genes, like *HBB*, PrediXcan and UTMOST both identified eQTLs predominantly in a single tissue and eQTL weights followed a normal distribution. (C) and (D) For genes that are differentially expressed in multiple tissue, like *APOE*, PrediXcan and UTMOST both estimated normally distributed eQTLs weights. However, UTMOST was able to identify more eQTLs that are functioning across tissues. (E) and (F) For genes that are ubiquitously expressed in all tissues, like USP40, eQTL weights estimated from PrediXcan and UTMOST were both normally distributed. And again, UTMOST was able to identify more eQTLs that are effective in more than one tissues.
(TIF)

**S4 Fig. Statistical difference of gene prioritization power of different TWAS methods.** Power was the proportion of successfully prioritized gene-trait associations in the causal tissue out of all associations under the same simulation setting. X-axis is the number of gene-expressing tissues. Each column stands for the proportion of eQTLs that are shared among tissues for a gene. Each row is the similarity of gene expression profiles across tissues which is estimated by correlation. Moving from the top left to the bottom right is a gradient spectrum from tissue-specific genes to broadly expressed genes. The colors represent different TWAS methods and y-axis is the power. For tissue-specific genes at the top left, single-tissue TWAS (Elastic Net-SLR) and cross-tissue TWAS (Group LASSO-GBJ) had similar power. For broadly expressed genes at the bottom right, cross-tissue TWAS (Group LASSO-GBJ) had greater power. The difference in power among different TWAS methods were statistically evaluated ([*] p-value < 0.05, [**] p-value < 0.01, [***] p-value < 0.0001).
(TIF)

**S5 Fig. TWAS gene prioritization power when $R^2_{expression-trait}$ = 0.5%.**
(TIF)

**S6 Fig. TWAS gene prioritization power when $R^2_{expression-trait}$ = 0.05%.**
(TIF)

**S7 Fig. TWAS gene prioritization power when $R^2_{expression-trait}$ = 0.001%.**
(TIF)

**S8 Fig. Statistical difference of type I error rates of different TWAS methods.** Type I error rate was the probability that TWAS wrongly identified a gene-trait association as significant while there was not any signal. Association p-values were controlled for the number of genes and tested tissues. X-axis is the number of gene-expressing tissues. Each column stands for the proportion of eQTLs that are shared among tissues for a gene. Each row is the similarity of gene expression profiles across tissues which is estimated by correlation. Moving from the top left to the bottom right is a gradient spectrum from tissue-specific genes to broadly expressed genes. The colors represent different TWAS methods and y-axis is the type I error rate. All TWAS methods had controlled type I error rates ($\leq$ 5%). The difference in type I error rates among different TWAS methods were statistically evaluated ([*] p-value < 0.05, [**] p-value < 0.01, [***] p-value < 0.0001).
(TIF)

**S9 Fig. Power when single-tissue associations were not adjusted for the number of tested tissues and $R^2_{expression-trait}$ = 1%.**
(TIF)

**S10 Fig. False positive rates when single-tissue associations were not adjusted for the number of tested tissues.**
(TIF)

**S11 Fig. Trait heritability estimation design and workflow.**
(TIF)

**S12 Fig. Estimation of trait heritability for simulated datasets.**
(TIF)

**S13 Fig. Tissue distribution of predicted GReX.** X-axis is the number of tissues that a gene was predicted to be expressed in based on GTEx v8 MASHR-based eQTL models. Y-axis is the count of genes. Genes tended to express in few numbers of tissue according to GTEx v8 prediction.
(TIF)

**S14 Fig. Empirical distribution simulated $E_{N×P}$ at five $cor(tissue_p, tissue_{p'})$ (0, 0.2, 0.4, 0.6, 0.8).** The number of tissues were five in this evaluation. Five thousand rounds of simulations were repeated for each value of $cor(tissue_p, tissue_{p'})$. In each repetition, we obtained one mean variance and one mean off-diagonal correlation across the five simulated tissues. (**A**) The empirical mean variance of $E_{N×P}$ were approximately one in any situations. The empirical mean off-diagonal correlations equaled 0, 0.186, 0.373, 0.563, and 0.756, for $cor(tissue_p, tissue_{p'})$ = 0 (**B**), 0.2 (**C**), 0.4(**D**), 0.6 (**E**), 0.8 (**F**).
(TIFF)

**S15 Fig. Distribution of locus RCP on the ACTG data.** To evaluate for an appropriate locus RCP cutoff, we inspected locus RCP statistics for colocalization tests of 49 tissues, 10 statistically significant phenotypes, and ~20K genes. Approximately 10M were present in the figure. Y-axis were cut out at count of 100 for the ease of visualization. We took locus RCP > 0.025 as a cut-off based on the locus RCP distribution.
(TIFF)

**S16 Fig. Alternative TWAS analytic framework based on simulation results.**
(TIF)

**S17 Fig. The advantage of our tissue specificity-aware TWAS analytic framework (top) in comparison to a regular single-tissue TWAS (bottom).** X-axis is the genomic location and y-axis is the -log10 transformed p-values. Colors denote different ACTG baseline laboratories. The significance threshold was $1.12×10^{-7}$ for both top and bottom TWAS frameworks. While single-tissue TWAS were able to find multiple significant gene-trait associations, the significant tissues did not necessarily connect to the traits of interest pathology-wise. Our tissue specificity-aware TWAS framework was able to retain all significant associations that were identified through regular single-tissue TWAS.
(TIF)

**S1 Table. Pairwise Comparison of power across all pairs of TWAS methods.**
(XLSX)

**S2 Table. Pairwise Comparison of type I error rates across all pairs of TWAS methods.**
(XLSX)

**S3 Table. Credible sets of baseline laboratory trait-related genes.**
(XLSX)

## Acknowledgments

The authors are grateful to the many persons living with HIV who volunteered for ACTG protocols A5095, A5142, ACTG 384, A5202, and A5257. In addition, they acknowledge the contributions of study teams and site staff for these protocols. We thank Paul J. McLaren, PhD (Public Health Agency of Canada, Winnipeg, Canada) for prior involvement and collaborations that used these genome-wide genotype data. Study drugs were provided by DuPont Pharmaceutical Company, Bristol-Myers Squibb, Inc., Agouron Pharmaceuticals, Inc., Glaxo-Wellcome, Inc., Merck and Co., Inc. Boehringer-Ingelheim Pharmaceuticals, Inc., Gilead Sciences, Inc., GlaxoSmithKline, Inc., Abbott Laboratories, Inc., Tibotec Therapeutics. The clinical trials were ACTG 384 (ClinicalTrials.gov: NCT00000919), A5095 (NCT00013520), A5142 (NCT00050895), A5202 (NCT00118898), and A5257 (NCT00811954). We thank Dr. Yiming Hu and Zhaolong Yu from Yale University and Alvaro Barbeira and Dr. Hae Kyung Im from the University of Chicago for technical support. We thank Dr. Shefali S. Verma and Anastasia Lucas from Dr. Marylyn Ritchie's lab at the University of Pennsylvania for suggestions and assistance in data visualization.

The content is solely the responsibility of the authors and does not necessarily represent the official views of the National Institute of Allergy and Infectious Diseases or the National Institutes of Health.

## Author Contributions

**Conceptualization:** Binglan Li, Marylyn D. Ritchie.

**Data curation:** Binglan Li, Anurag Verma, Yuki Bradford, David W. Haas.

**Formal analysis:** Binglan Li.

**Funding acquisition:** David W. Haas, Marylyn D. Ritchie.

**Investigation:** Binglan Li, Marylyn D. Ritchie.

**Methodology:** Binglan Li, Yogasudha Veturi.

**Resources:** Eric S. Daar, Roy M. Gulick, Sharon A. Riddler, Gregory K. Robbins, Jeffrey L. Lennox, David W. Haas, Marylyn D. Ritchie.

**Software:** Binglan Li.

**Supervision:** Marylyn D. Ritchie.

**Visualization:** Binglan Li.

**Writing – original draft:** Binglan Li.

**Writing – review & editing:** Yogasudha Veturi, Anurag Verma, Yuki Bradford, Eric S. Daar, Roy M. Gulick, Sharon A. Riddler, Gregory K. Robbins, Jeffrey L. Lennox, David W. Haas, Marylyn D. Ritchie.

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
