## [Decision Letter · Decision Letter 0]

10 Nov 2020

Dear Dr Ritchie,

Thank you very much for submitting your Research Article entitled 'Tissue specificity-aware TWAS (TSA-TWAS) framework identifies novel associations with metabolic, immunologic, and virologic traits in HIV-positive adults' to PLOS Genetics. Your manuscript was fully evaluated at the editorial level and by independent peer reviewers. The reviewers appreciated the attention to an important problem, but raised some substantial concerns about the current manuscript. Based on the reviews, we will not be able to accept this version of the manuscript, but we would be willing to review again a much-revised version. We cannot, of course, promise publication at that time.

If you decide to revise the manuscript for further consideration at PLOS Genetics, please aim to resubmit within the next 60 days, unless it will take extra time to address the concerns of the reviewers, in which case we would appreciate an expected resubmission date by email to plosgenetics@plos.org.

[LINK]

We are sorry that we cannot be more positive about your manuscript at this stage. Please do not hesitate to contact us if you have any concerns or questions.

Yours sincerely,

John S Witte

Guest Editor

PLOS Genetics

Hua Tang

Section Editor: Natural Variation

PLOS Genetics

Reviewer's Responses to Questions

**Comments to the Authors:**

Reviewer #1: Li et al. proposed a novel framework to integrate and maximize the performance of multiple TWAS methods.

1. In Figure 2, why the statistical power decreases with the gene expressing tissues increase? Do those genes have the same genetic heritability in different scenarios?

2. In line 265-266, the authors claimed that “if trait-related tissue(s) are uncertain, it may be better to stratify genes based on the number of tissues in which the genes are predicted to be expressed”, but it is not clear how the number of tissues can affect the choice of TWAS frameworks.

3. The authors claimed that the novel framework considers the potential false positive rates of tissue identifications. In real-data analysis of 37 baseline laboratory values, authors didn’t show the tissue-trait association results. At least, for those identified trait-associated genes, it is not clear in which tissues those genes were identified.

4. In line 323 and 324, the authors claimed that some of those pleotropic genes were located on the MHC region of chromosome 6, while the MHC region has a rather complicated LD structure and thus those identified results could possibly be false positives. However, the authors didn’t discuss about this point.

5. In line 431, “Genotypes were simulated as biallelic SNPs”, were LD structures considered when simulating genotypes?

6. To show the power/false discovery rate of the proposed framework, it will be better if the author can conduct simulation analysis using the proposed framework or at least summarize the potential performance of the proposed framework based on the current simulation results.

Reviewer #2: Binglan Li and colleagues present a comprehensive evaluation of single-tissue and cross-tissue transcriptome-wide association study methods using simulations with a focus on two methods PrediXcan and UTMOST. They develop a tissue-specificity aware analytic framework and apply their results GWAS data for laboratory traits from AIDS clinical trials. Overall this is an important methodological contribution to the growing field of transcriptome-wide association studies that helps further clarify the role of tissue specificity and cross-tissue and pleiotropic effects in influencing TWAS results. I do however have a few concerns and these are elaborated on below.

Major comments:

In their application of the TSA-TWAS framework to the ACTG data set, the authors identify several gene expression-trait associations where for the same trait multiple genes in the same genomic region are implicated. For example, UGT1A6-MROH2A-UGT1A1-UGT1A7 and total bilirubin in Table 3 and CD2AP and RP11-385F7.1 and absolute basophil count in Table 4, to name a couple of such instances. The current implementations of PrediXcan and its variations recommend coupling the analysis with enloc (reference 57 in this manuscript by Li et al). Why do the authors choose not to evaluate colocalisation between the GWAS signal and the eQTL signal to potentially filter associations at such multi-gene association regions?

Page 26, line 515, under Phenotype: How was linkage disequilibrium (LD) structure modelled, if at all, in the GWAS data set for the Phenotype in the simulations? There seem to be two forms of correlations that affect TWAS results that aren't clearly addressed in the simulation -- (1) correlation between SNPs due to LD in the phenotype or trait GWAS and (2) expression correlation between genes that are located physically near each other on the genome (a known phenomenon, see for example, Michalak, Genomics, Volume 91, Issue 3, March 2008, Pages 243-248). The authors highlight UGT1A1 as the target gene in the Abstract but rightly suggest the potential for multi-gene involvement on page 19, line 363 but this presents a paradox that is not addressed. What is the evidence from GWAS that there is a single causal gene at a locus or multiple causal genes at a locus for complex traits? How is this accounted for in TSA-TWAS and its underlying simulation framework? How does this affect the power of the approaches?

Page 18, line 318: are the pleiotropic genes expressed across tissues or are they tissue specific?

Minor comment:

Page 15, lines 311 to 313 (ATF6B...) - "For instance, ATF6B is a ... and it has been associated

with HIV-associated neurocognitive disorders in previous research." - please provide a citation to support this statement.

**Have all data underlying the figures and results presented in the manuscript been provided?**

Reviewer #1: None

Reviewer #2: Yes

PLOS authors have the option to publish the peer review history of their article (what does this mean?). If published, this will include your full peer review and any attached files.

Reviewer #1: No

Reviewer #2: No

---

## [Decision Letter · Decision Letter 1]

2 Feb 2021

Dear Dr Ritchie,

Thank you very much for submitting your Research Article entitled 'Tissue specificity-aware TWAS (TSA-TWAS) framework identifies novel associations with metabolic, immunologic, and virologic traits in HIV-positive adults' to PLOS Genetics.

The manuscript was fully evaluated at the editorial level and by independent peer reviewers. The reviewers appreciated the attention to an important topic but identified some concerns that we ask you address in a revised manuscript

We therefore ask you to modify the manuscript according to the review recommendations. Your revisions should address the specific points made by each reviewer.

[LINK]

Yours sincerely,

John S Witte

Guest Editor

PLOS Genetics

Hua Tang

Section Editor: Natural Variation

PLOS Genetics

Reviewer's Responses to Questions

**Comments to the Authors:**

Reviewer #1: The authors have addressed most of previous concerns, but some still need clarifications:

1. In responses to “In Figure 2, why the statistical power decreases with the gene expressing tissues increase? Do those genes have the same genetic heritability in different scenarios?”, the author claimed that the decreasing power is due to the increased number of tests, but it shouldn’t be case for the Group Lasso-GBJ since it is a burden test considering all tissues together. This figure is also contradictory to Line 173-175

2. In Figure 6, Group-Lasso GBJ showed comparably lower power than other methods when a higher percentage of genes are tissue-specific genes, while in Fig 2, all methods have lower power for genes expressed in multiple tissues. It is understandable for Group-Lasso GBJ to have higher type-I error rates for tissue-specific genes, but it will be better to discuss about the lower power of Group-Lasso GBJ method

Reviewer #2: Thank you for the clear and thoughtful responses and for revising the manuscript.

**Have all data underlying the figures and results presented in the manuscript been provided?**

Reviewer #1: None

Reviewer #2: Yes

PLOS authors have the option to publish the peer review history of their article (what does this mean?). If published, this will include your full peer review and any attached files.

Reviewer #1: No

Reviewer #2: **Yes: **Siddhartha Kar

---

## [Editor Report · Decision Letter 2]

3 Mar 2021

Dear Dr Ritchie,

We are pleased to inform you that your manuscript entitled "Tissue specificity-aware TWAS (TSA-TWAS) framework identifies novel associations with metabolic, immunologic, and virologic traits in HIV-positive adults" has been editorially accepted for publication in PLOS Genetics. Congratulations!

Yours sincerely,

John S Witte

Guest Editor

PLOS Genetics

Hua Tang

Section Editor: Natural Variation

PLOS Genetics

Comments from the reviewers (if applicable):

**Data Deposition**

http://datadryad.org/submit?journalID=pgenetics&manu=PGENETICS-D-20-01385R2

**Press Queries**

---

## [Editor Report · Acceptance letter]

21 Apr 2021

PGENETICS-D-20-01385R2 

Tissue specificity-aware TWAS (TSA-TWAS) framework identifies novel associations with metabolic, immunologic, and virologic traits in HIV-positive adults 

Dear Dr Ritchie, 

We are pleased to inform you that your manuscript entitled "Tissue specificity-aware TWAS (TSA-TWAS) framework identifies novel associations with metabolic, immunologic, and virologic traits in HIV-positive adults" has been formally accepted for publication in PLOS Genetics! Your manuscript is now with our production department and you will be notified of the publication date in due course.

With kind regards,

Katalin Szabo

PLOS Genetics

On behalf of:
